Taxonomic and systematic revisions to the North American Nimravidae (Mammalia, Carnivora)

Barrett Paul Z. pzbarr@gmail.com
Department of Geology and Geologic Engineering, South Dakota School of Mines and Technology , Rapid City, SD , United States of America
De Baets Kenneth
Electronic publication date: 2016 Feb 9
Publication date: 2016
Volume: 4
Electronic Location ID: e1658
Received 2015 Sep 10; Accepted 2016 Jan 13
Copyright: ©2016 Barrett
Copyright year: 2016
Copyright holder: Barrett
License: This is an open access article distributed under the terms of the Creative Commons Attribution License, which permits unrestricted use, distribution, reproduction and adaptation in any medium and for any purpose provided that it is properly attributed. For attribution, the original author(s), title, publication source (PeerJ) and either DOI or URL of the article must be cited.
License URL: https://creativecommons.org/licenses/by/4.0/

Keywords: Nimravidae, Eusmilus, Taxonomy, Hoplophoneus oharrai, Systematics, Pogonodon, Dinictis

Funding: The author received no funding for this work.

==============================
The Nimravidae is a family of extinct carnivores commonly referred to as “false saber-tooth cats.” Since their initial discovery, they have prompted difficulty in taxonomic assignments and number of valid species. Past revisions have only examined a handful of genera, while recent advances in cladistic and morphometric analyses have granted us additional avenues to answering questions regarding our understanding of valid nimravid taxa and their phylogenetic relationships. To resolve issues of specific validity, the phylogenetic species concept (PSC) was utilized to maintain consistency in diagnosing valid species, while simultaneously employing character and linear morphometric analyses for confirming the validity of taxa. Determined valid species and taxonomically informative characters were then employed in two differential cladistic analyses to create competing hypotheses of interspecific relationships. The results suggest the validity of twelve species and six monophyletic genera. The first in depth reviews of Pogonodon and Dinictis returned two valid species (P. platycopis, P. davisi) for the former, while only one for the latter (D. felina). The taxonomic validity of Nanosmilus is upheld. Two main clades with substantial support were returned for all cladistic analyses, the Hoplophoneini and Nimravini, with ambiguous positions relative to these main clades for the European taxa: Eofelis, Dinailurictis bonali, and Quercylurus major; and the North American taxa Dinictis and Pogonodon. Eusmilus is determined to represent a non-valid genus for North American taxa, suggesting non-validity for Old World nimravid species as well. Finally, Hoplophoneus mentalis is found to be a junior synonym of Hoplophoneus primaevus, while the validity of Hoplophoneus oharrai is reinstated.

Introduction

The Nimravidae is a family of extinct, superficially “cat-like” carnivores, most of which exhibit saber-tooth dentition. They existed from the late Eocene to the end of the Oligocene (approximately 35.5–23.0 Ma) and are known from North America, Europe and Asia (Bryant, 1996; Peigné, 2003).

The first nimravid specimen was described in 1851 and by 1992 all currently recognized North American nimravid taxa were named. However, throughout the course of this group’s taxonomic history there has been little semblance of stability insofar as specific or generic diagnoses. Much of the specific confusion for this group arose from the non-explicit species concepts and diagnosing criteria utilized by authors. However, it can be gleaned that in the taxonomic history of North American nimravids, species diagnoses were often based on minor metrical differences between specimens, even though this would be expected to vary somewhat in all species. For example, Hoplophoneus robustus “…was proposed as representative of [a second, at the time]… larger skull, which was referred by Leidy to H. primaevus. As compared with that species, it shows an increase in size and the skeleton is more massive” (Adams, 1896b, p. 428) Pogonodon cismontanus was diagnosed on the criteria of “The size is close to that of the type genus, P. platycopis Cope, although in nearly every dimension this specimen is slightly smaller” (Thorpe, 1920, p. 223). Additionally, Hoplophoneus belli was named for defining characters such as “Size approximating that of Hoplophoneus cerebralis Cope from the John Day, but smaller. Superior canine distinctly more slender than H. cerebralis or in H. oreodontis” (Stock, 1933, p. 37).

From this air of general confusion several major taxonomic revisions were undertaken in the middle part of the 20th century (Hough, 1949; Morea, 1975; Simpson, 1941; Toohey, 1959). Most of this work focused on the Hoplophoneus and Eusmilus genera which have historically been difficult to distinguish, and contained what was perceived as a superfluous number of species (Jepsen, 1933; Scott & Jepsen, 1936; Sinclair, 1924). Both Simpson (1941) and Hough (1949) were primarily concerned with specimens originating from the Brule Formation of South Dakota, and adjoining states. This limited geographic and temporal sample was hoped to remove misleading morphological signals of regional variation and evolutionary trends. Likewise, Toohey (1959) reviewed the then eight Nimravus species present in the literature to find clear diagnosable criteria, though expanded his sample to include all North American specimens due to the rarity of this taxon. Finally, Morea (1975) assessed the generic assignment and taxonomic validity of Eusmilus in relation to Hoplophoneus species utilizing all described North American material.

All of these prior taxonomic revisions implicitly utilized a variant of a morphological species concept, whereby species were recognized under morphological clusters of size and shape similarity. More specifically, their operating hypotheses were: uniform possession of certain morphologic features equated to a valid species, variance within this species was otherwise dimensional (metrical) and interpreted as tokogenetic, e.g., sexual dimorphism. To test these hypotheses, comparisons to extant felid taxa were made, typically at the subspecies level to mitigate regional variation. If ranges in size and shape were comparable between the extant taxa and nimravid specimens, then the hypothesis was upheld.

This methodology proves problematic for several reasons. First, without a defined species concept, consistency in describing the appropriate taxonomic level becomes uncertain. Since no standard is being followed, genera, sub-genera or intraspecific variation may end up being inadvertently diagnosed. Second, justification for what were the important morphological features and what was variable was never made. For example, Simpson (1941), states that the P2 is variable and taxonomically uninformative by referring to the work of Adams (1896b). However, in the latter work no analyses were performed, and the inference of the variability of the P2 was unjustified and simply said to coincide with age. Lastly, no statistical tests were performed to gauge the similarity of extant groups to nimravids, nor between nimravid taxa (a notable exception is Hough, 1949, though only between her two proposed subspecies of H. primaevus). These judgments of similarity were instead based on comparisons of raw values. In short, the shortcomings of these previous revisions might stem from a non-explicit species concept, lack of character criteria, and subjective assessments of similarity.

Genus and tribe-level taxonomy

Relationships of nimravids at the specific level and above have also garnered much debate. Traditionally, perceived evolutionary lineages were organized into tribes based on determined trends in morphology (e.g., De Beaumont, 1964; Martin, 1980) or recovered relationships in cladistic analyses (Bryant, 1996; Peigné, 2003) (Table 1).

Table 1 Nimravid tribes.

Previous hypotheses of nimravid lineages (tribes) with representative genera.

De Beaumont (1964)	
Nimravini	Dinictini	Hoplophoneini	Eusmilini	
Nimravus	Dinictis	Hoplophoneus	Eusmilus	
Dinaelurus		Sansanosmilus		
Dinailurictis				
Martin (1980)	
Nimravini	Dinaelurini	Hoplophoneini	Eusmilini	Barbourofelini	
Nimravus	Dinaelurus	Hoplophoneus	Eusmilus	Barbourofelis	
Dinictis					
Pogonodon					
Bryant (1996)	Peigné (2003)	
Nimravini	Hoplophoneini	Nimravini	Hoplophoneini	
Nimravus	Hoplophoneus	Nimravus	Nanosmilus	
Dinaelurus	Eusmilus	Dinaelurus	Hoplophoneus	
			Eusmilus	

Two cladistic analyses have been performed on nimravid taxa, Bryant’s 1996 study examined relationships solely within the North American members of the Nimravidae, sensu stricto, (eleven taxa) in a parsimony-based, implicit enumeration analysis utilizing twenty-seven characters. Bryant (1996) sought to preliminarily revise the taxonomy of the North American nimravids and subsequently generate a hypothesis of relationships through a cladistic analysis. Validation for species was based on the previous taxonomic revisions already discussed (Hough, 1949; Morea, 1975; Simpson, 1941; Toohey, 1959) and preliminary reconsiderations of the taxa Dinictis and Pogonodon. However, Bryant (1996) admits that the revised diagnoses may not “…comply completely with cladistics principles” (p. 454) and descriptive features were taken into consideration.

Character polarity was set by a hypothetical ancestor derived from selection of what constituted plesiomorphic (ancestral) character states in real taxa, such as the members of the Barbourofelinae, Aeluroidea, and Caniformia. Additionally, support metrics were calculated (in the form of nonparametric bootstrap analysis) for the resultant tree topology. Two nodes garnered over 90% support, partially dividing the North American nimravids into two tribes, the Nimravini and Hoplophoneini (Fig. 1). Every other node was described as less than 80% supported, and since these clades have historically been unnamed, they received no new designation.

Figure 1 Bryant (1996) nimravid phylogeny.

Recovered phylogenetic hypothesis of Bryant’s 1996 North American Nimravidae analysis. Numbers at nodes indicate tribal designation: (1), Hoplophoneini; (2), Nimravini. Modified from Bryant (1996: Fig. 1).

Figure 2 Peigné (2003) nimravid phylogeny.

Recovered phylogenetic hypothesis of Peigné (2003). Modified from Peigné (2003: Fig. 14).

Peigné’s 2003 study utilized eighteen taxa, of which one was a single specimen, and thirty-three characters in a parsimony-based implicit enumeration analysis similar to Bryant (1996). Peigné (2003) sought to systematically revise the European nimravid taxa and subsequently interpret their relationships with their North American relatives utilizing a cladistic analysis. European taxa underwent statistical linear morphometric and comparative analyses to determine species validity, while valid North American species, and their diagnoses, were derived from previous taxonomic investigations (Bryant, 1996; Martin, 1980; Martin, 1992; Toohey, 1959).

Similar to the methods of Bryant (1996) and Peigné (2003) utilized a hypothetical outgroup derived from selection of ancestral states in multiple real taxa to determine character polarity, but also considered ontogenetic information. This information was applied to two analyses, one consisting of eighteen taxa, and a second with the removal of three of the prior analysis’ taxa. A strict consensus tree was employed to summarize relationships of the three input trees retained in the second analysis (Fig. 2).

Study goals

No revision of nimravid taxa has directly examined the validity of the Dinictis, Pogonodon, and Nanosmilus genera and their contained species through cladistic, morphometric, or other analytical techniques. Additionally, numerous advancements in morphometric and cladistic analyses have taken place in the last twenty years allowing potentially more informed views into the specific and generic validity and evolutionary history of this group. The aim of this study was to (1) revise the taxonomy of North American nimravids, specifically that of valid species, and (2) determine the validity of North American genera and the phylogenetic relationships New and Old World nimravid taxa share. To accomplish the first goal, a defined species concept and accompanying character criteria were employed, along with statistical analyses and exploratory techniques. The second goal was facilitated via cladistic analyses utilizing both parsimony and posterior probability optimality criteria with hopes that agreement between the two would supply additional support on the generic allocation of species, not to mention hypothesis of evolutionary history.

Materials & Methods

Species concept

To maintain consistency in diagnosing valid species of nimravid taxa, a defined species concept was strictly followed for this study. Specifically, the species concept utilized was the phylogenetic species concept (PSC), where a species equates to “…the smallest aggregation of populations (sexual) or lineages (asexual) diagnosable by a unique combination of character states in comparable individuals (semaphoronts)” (Nixon & Wheeler, 1990, p. 218). The PSC is a character-based and pattern-based species concept which eliminates dependence on particular kinds of processes, such as reproductive isolation. Even so, the PSC is consistent with numerous processes potentially responsible for speciation, such as sympatric or allopatric speciation (Nixon & Wheeler, 1990). Additionally, by using the PSC one can readily perform a cladistic analysis without deriving a new set of characters, advantageous for the second goal of this study.

However, when applying this definition to a fossil sample, shortcomings become immediately evident. The PSC speaks of biological populations when diagnosing sexual organisms, though often in the fossil record, no arguably preserved populations can be found. Such is the case with nimravids, which being top predators, were rare in their ecosystems, leaving us with only isolated specimens, not biological populations. In light of this issue with the fossil record, and due to the PSC’s character-based definition, the identification of a valid species is better viewed as the determination of valid characters, i.e., morphological features which denote species or higher taxonomic-level designation.

Since it is impossible, without an a priori assumption, to determine what morphological features on a specimen equate to characters, and not traits (here meaning variation within a species such as sexual dimorphism or tokogeny), all observed variation between specimens was initially entertained. This approach allowed the assured inclusion of all characters, but necessarily that of all traits as well. To remove the signature of uninformative traits from the final character list, character criteria were employed and morphometric analyses conducted, creating the general procedure utilized by this study as outlined in Fig. 3. More specifically, two sets of data were collected from nimravid specimens: (1) linear measurements; and (2) a list of morphologic variability which could be broken into discrete character states. From the former dataset, cluster analyses were performed to find natural groupings which were then tested for support via discriminant function analysis (DFA). The resultant supported morphogroups were then compared to a character list filtered by specified criteria. Universally shared character states per morphogroup then formed the basis of this study’s PSC species. This procedure garners a substantially more objective approach to the determination of what are useful taxonomic features for this group since they are based upon statistically significant relationships within a separate linear morphometric dataset and less onindividual author’s subjective taxonomic opinion.

Figure 3 Study methods.

Diagrammatic representation of the procedure utilized to determine valid species for this study. Steps within boxes represent analytical techniques employed to determine character validity.

Figure 4 Basilar length.

Measured as the prosthion (pr) to the basion (ba) in (mm). 1(0): 105 and smaller; 1(1): 110–200; 1(2): 205 and larger. Figured specimen UNSM 322-51, Hoplophoneus occidentalis.

Figure 5 Lacrimal and jugal suture.

Sutural contact between the lacrimal (Lac) and jugal (Ju). 2(0): present (A); 2(1): absent (B). Figured specimens YPM 10045, Nimravus brachyops (A), USNM 18214, Hoplophoneus sp. (B).

Figure 6 Masseteric fossa on the lateral surface of the maxilla and jugal.

3(0): shallow or absent (A); 3(1): deep with distinct dorsal margin (B). Figured specimens UNSM 1068, Hoplophoneus dakotensis (A), AMNH 6930, Nimravus brachyops (B).

Character criteria

Due to the specified constraints of the PSC and operational necessities of cladistic analysis, characters needed to be restricted in certain ways to be utilized by this study. Nixon & Wheeler (1990) were explicit in the nature of characters utilized in diagnosing species, so to maintain consistency those criteria were strictly followed. Specifically, characters and their associated states were required to be of the same quality as those used in cladistic analysis while additionally being only parsimony-informative (i.e., neither autapomorphic nor polymorphic for a taxon). This decision was further justified by the latter constraint of performing cladistic analyses on the same character set to infer evolutionary history. To facilitate interpretive power of these analyses, only characters which could be coerced into discrete states, and were additionally justifiably independent were entertained. Together, these decisions formed the basis of the character criteria employed for this study.

Figure 7 Medial fossa zygomatic arch.

Fossa on the medial face of the zygomatic arch, below the postorbital process. 4(0): no fossa (A); 4(1): presence of a marked fossa (B). Figured specimens UNSM 1068 (reversed), Hoplophoneus dakotensis (A), AMNH 6931, Nimravus brachyops (B).

Figure 8 Zygomata shape in dorsal view.

5(0): broadly circular (A); 5(1): triangular (B). Figured specimens YPM 10045, Nimravus brachyops (A), UNSM 1072, Hoplophoneus primaevus (B).

Character analysis

Sources for morphologic variation and potential characters originated from a variety of publications, specifically, previous nimravid cladistic studies (Bryant, 1996; Peigné, 2003), descriptive studies of nimravid morphology (Boyd & Welsh, 2013; Bryant, 1988; Hough, 1949; Morea, 1975), original observations, and recent felid morphological cladistic analyses (King, 2012; Spearing, 2013). This latter source was primarily entertained as inspiration for investigation, as it would seem likely that the highly convergent nature nimravids share with the feliform body plan at least some aspects of their morphology would co-vary.

Character scoring

Most of the reported North American nimravid material was examined and scored for this study (191 specimens), see Appendix S1. Specimens examined originated from the following institutions: the Museum of Geology at South Dakota School of Mines and Technology, the University of Nebraska Lincoln State Museum, the American Museum of National History, the Yale Peabody Museum of Natural History, the United States National Museum, and Badlands National Park. The whereabouts of the holotype, and only known specimen, of Dinaelurus crassus is currently unknown, and was thus unable to be examined in person. Therefore, this taxon was scored from literature (Bryant, 1996; Eaton, 1922; Peigné, 2003).

Figure 9 Medial ridge on palate.

6(0): absent (A); 6(1): present (B). Figured specimens UNSM 25512, Dinictis felina (A), UNSM 322-51, Hoplophoneus occidentalis (B).

Figure 10 Basicranial foramina.

Relationship between petrobasilar (pbf) and posterior lacerate (plf) foramina. 7(0): foramina confluent (A); 7(1): two discrete foramina present (B); 7(2): petrobasilar and posterior lacerate form two distinct grooves (C). Figured Specimens FAM 83386, Daphoenodon sp. (A), UNSM 279-51, Nimravus brachyops (B), UNSM 25506, Hoplophoneus primaevus (C).

Tools utilized

Specimens were scored by hand or from the literature when necessary. To score specimens some potential characters only required visual inspection, though others were aided through the use of the following instruments:

∙ 15 cm stainless steel digital calipers with 0.02 mm accuracy

∙ 1.5 m non-stretch fiberglass measuring tape, considered accurate to the nearest millimeter

∙ Stainless steel digital protractor with 0.3 degree accuracy

∙ Binocular microscope with 50 × magnification

Employing character criteria

To bring potential characters in line with the requirements of the PSC and cladistic analyses, several methods were employed. The first removed any character which displayed as polymorphic for a given specimen (i.e., displaying multiple character states, such as the presence of P1 on the right maxilla, but absence on the left), and thus was unaligned with the PSC which requires character states to be shared universally for a given taxon (Nixon & Wheeler, 1990). Additionally, characters were removed if: they showed non-variation for all examined specimens, were unable to be consistently scored due to vague descriptions and landmarks presented in literature, or lacked natural gaps for continuous characters. The resulting revised character list was then used in tandem with the results of the subsequently described morphometric analyses.

Figure 11 Postglenoid foramen (pgf).

8(0): present (A); 8(1): absent (B). Figured specimens UNSM 25512, Dinictis felina (A), UNSM 322-51, Hoplophoneus occidentalis (B).

Figure 12 Mastoid and paroccipital morphology.

Morphology of mastoid (Ma) and paroccipital (Pop) processes. 9(0): reduced mastoid with large plate-like paroccipital process (A); 9(1): large tabular mastoid with reduced to near absent paroccipital process (B). Figured specimens YPM 10045, Nimravus brachyops (A) USNM 18214, Hoplophoneus sp. (B).

Morphometric analyses

To quantify the metric differences in size and shape of the examined specimens, and by extension find statistically supported natural groupings, fifteen standard (Boyd & Welsh, 2013; Bryant, 1988; Hough, 1949; Morea, 1975; Simpson, 1941; Toohey, 1959) measurements (Fig. S2) were taken (when possible) on all specimens examined during the study, utilizing the instruments previously listed. The resulting metric dataset can be seen in Appendix S3.

All analyses were conducted in R, version 3.2.2 (2015-08-14) (R Core Team, 2015), with the aid of R Studio, version 0.99.467 (RStudio Team, 2015). A total of four sets of analyses were undertaken at the generic level to mitigate issues with the below described imputation procedures, while utilizing at their core cluster and Discriminant Function Analysis (DFA).

Figure 13 Glenoid socket.

Shape of the glenoid socket. 10(0): anterior lip is missing (A); 10(1): Posterior lip of the glenoid socket projects more ventrally than anterior lip (B); 10(2): anterior lip and posterior lip project equally ventrally (C). Figured specimens AMNH 62026, Pogonodon davisi (A), UNSM 322-51, Hoplophoneus occidentalis (B), AMNH 6941 (reversed), Hoplophoneus cerebralis (C).

Figure 14 Braincase angle.

Angle between the braincase, disregarding the sagittal crest, and the axial plane of the cranium. 11(0): oblique (A); 11(1): parallel, or nearly so (B). Figured specimens UNSM 1072, Hoplophoneus primaevus (A), AMNH 6941 (reversed), Hoplophoneus cerebralis (B).

Datasets were initially modified to remove variables or specimens which exhibited superfluous amounts of missing data. For the remainder of the relatively complete datasets, imputation was employed to ready them for multivariate analyses, which require no missing data. To that end, the R package “norm” version 1.0-9.5 (Novo & Schafer, 2013) was utilized due to its previously determined robustness and accuracy for morphometric datasets (Clavel, Merceron & Escarguel, 2014). The norm package operates via a multiple imputation (MI), data augmentation (DA) approach, assuming a joint multivariate normal distribution. For this study, 100 data augmentation iterations (steps) were employed for each dataset imputed, as suggested by Clavel, Merceron & Escarguel (2014).

Table 2 Character sources.

Sources of characters utilized for this study. If a character was derived from a previous cladistic analysis, it is cited with a reference and corresponding character number in its respective columns. Corresponding characters across other cladistic studies are also presented in their respective columns. King (2012) utilized multiple character lists, depending on outgroup, the letter designation next to each of that study’s characters equates to the appendix in which it can be found.

Character	Derived from	Bryant, 1996	Peigné, 2003	King, 2012	Spearing, 2013	Character type	
1	Original character					Cranial	
2	Bryant, 1996	1	8			Cranial	
3	Bryant, 1996	2				Cranial	
4	Peigné, 2003	3	12			Cranial	
5	Original character					Cranial	
6	Original character					Cranial	
7	Peigné, 2003		14			Cranial	
8	Bryant, 1996	6				Cranial	
9	Peigné, 2003	7	1	17 B, 38 B	11, 18	Cranial	
10	Spearing, 2013				4	Cranial	
11	Morea, 1975 (p. 29)					Cranial	
12	Bryant, 1988 (p. 109)					Cranial	
13	Original character					Mandibular	
14	Spearing, 2013				66	Mandibular	
15	Peigné, 2003		2		65	Mandibular	
16	Peigné, 2003		3		68	Mandibular	
17	Peigné, 2003		15			Mandibular	
18	Peigné, 2003		7		134, 135	Dental	
19	Bryant, 1996	11				Dental	
20	Boyd & Welsh, 2013				127	Dental	
21	Peigné, 2003	14	19		77	Dental	
22	Bryant, 1996	16				Dental	
23	Peigné, 2003	17	22			Dental	
24	Peigné, 2003	20	24			Dental	
25	Peigné, 2003		6		79	Dental	
26	Peigné, 2003	15	28		96	Dental	
27	Peigné, 2003		29			Dental	
28	Peigné, 2003	18	30			Dental	
29	Peigné, 2003	21	31		102	Dental	
30	Peigné, 2003	22	32		103	Dental	
31	Spearing, 2013	24			125	Dental	
32	Bryant, 1996	26			181	Postcranial	
33	Bryant, 1996	27				Postcranial	

Multivariate normality was then checked for all datasets (a condition assumed for DFA) via the R package “ICS” version 1.2-5 (Nordhausen, Oja & Tyler, 2008). The test specifically employed was the multivariate normality test based on kurtosis.

Following the aforementioned preparation of the data, cluster analysis was performed on the dataset corresponding to each genus utilizing a UPGMA clustering algorithm. From the resultant dendrogram, differential sets of group assignment were entertained for discriminant function analysis. For all DFAs, datasets (save the Pogonodon set) were split into equal training and testing sets which were additionally designed to retain the proportional makeup of the data as a whole for each set with the aid of the R package “caret” version 6.0-58 (Kuhn et al., 2015). For the DFAs, the R package “MASS” version 7.3-43 (Venables & Ripley, 2002) was utilized to generate a model of discrimination for the training set with probabilities set equally likely that a specimen would belong to either group to account for differences in sample size. The variable coefficients for the respective DFAs can be seen in Appendix S4. From this DFA created model, predictions were made of group assignment accuracy when the test set was analyzed. Ninety percent was chosen as the cutoff for morphometric distinction between groups for all discriminant function analyses, as suggested by Hammer & Harper (2006), p. 96. This specific value was somewhat arbitrary since even 100 percent discrimination could represent a myriad of relationships beyond species delineation, such as sexual dimorphism, regional or temporal variation. Instead, the cutoff value was employed as a way to reduce the number of morphogroups that needed to be examined (while still having substantial support), and valid species identification relied upon the presence or absence of a set of shared unique character state combinations for a given morphogroup. The R code and specific datasets used to perform these analyses can be seen in Appendices S5 and S6, respectively.

Figure 15 Postorbital process (Po) of frontal.

12(0): projects horizontally (A); 12(1): projects ventrally (B). Figured specimens YPM 10045, Nimravus brachyops (A), AMNH 39101, Hoplophoneus primaevus (B).

Figure 16 Anterior mandible position.

13(0): in line with the tooth row, mandibular border of cheek tooth row is in the same plane as the mandibular border of the incisors and canines (A); 13(1): elevated above tooth row (B); 13(2): cheek teeth and anterior teeth brought again into same plane by elevation of cheek teeth on pedestal (C). Figured specimens YPM 10066, Daphoenus vetus (A), YPM 10045, Nimravus brachyops (B), YPM PU 11079, Hoplophoneus dakotensis (C), modified from Hatcher (1895: Plate XL).

Pogonodon

The Pogonodon dataset faced a unique problem in having a high level of incompleteness, further complicated by a lack of overlapping morphology amongst specimens. To remedy this situation, the most common structural element (the dentary, with its four associated variables) was chosen as the basis for morphological comparisons. This limited number of variables was also required due to the inability of running a discriminant function analysis with a greater number of variables than cases in the input matrix. Doing so would result in singular variance/covariance matrix which would be unable to be computed. Thus a hard limit was placed upon the number of variables analyzed in the cluster and discriminant function analyzes for this genus.

Likewise to the other genera, group assignment was initially determined via cluster analysis utilizing a UPGMA clustering algorithm. From the resultant dendrogram, the two most-inclusive tiers of group assignment were entertained (2 and 3 groups respectively) for differential DFAs. Training and testing sets were unable to be created for the DFAs due to low group membership (≤3 specimens for certain groups), which results in these groups not being proportional represented in both training or testing sets at best, or at worst not represented at all. Therefore, DFA was performed on the entirety of the Pogonodon dataset and jackknife cross-validation was utilized to gauge the accuracy of group assignment.

Figure 17 Coronoid process.

Development and orientation of the coronoid process (cp). 14(0): posteriorly orientated posterior border (A); 14(1): vertically orientated posterior border (B); 14(2): anteriorly orientated posterior border (C). Figured specimens UNSM 2509-59, Pogonodon platycopis (A), AMNH 39101, Hoplophoneus primaevus (B), PU 12953, Hoplophoneus sicarius (C), modified from Scott & Jepsen (1936: Plate XX).

Figure 18 Genial flange.

Size of the genial flange in adult taxa. Measured as the height of the genial flange from the anterior portion of the postcanine diastema to the ventral apex of the genial flange/length of dentary from the posterior articular surface to the most anterior aspect. 15(0): no flange, the ventral rim of the chin is regularly curved (A); 15(1): no flange, but the ventral rim of the chin is distinctly angulate (B); 15(2): short flange, between 22 and 31% of the total length of mandible (C); 15(3): deep flange, 32–50% of the total length of the mandible (D); 15(4): extremely deep flange, 54% or more of the total mandibular length (E). Figured specimens YPM 10066, Daphoenus vetus (A), YPM 10045, Nimravus brachyops (B), YPM PU 13587, Dinictis felina (C), YPM PU 11079, Hoplophoneus dakotensis (D), modified from Hatcher (1895: Plate XL), FAM 69377, Hoplophoneus cerebralis (E).

Valid species and characters

From the aforementioned character and morphometric analyses the final list of valid species and taxonomically informative characters were determined. Character analysis resulted in nine novel characters, while the remaining were modified from previous nimravid cladistic analyses (Bryant, 1996; Peigné, 2003). The source and correspondence of these characters across studies can be seen in Table 2. These characters are subsequently presented below (Figs. 4–36). Dental terminology follows the conventions utilized by prior nimravid authors (e.g., Bryant, 1996; Peigné, 2003), where upper dention is denoted with upper case letters (e.g., M1) and lower dentition with lower case letters (e.g., m1).

Cladistic analyses

Ingroup taxa

For this study, ingroup taxa originated from the results of the valid species analyses and include: Hoplophoneus primaevus (Leidy, 1851), H. occidentalis (Leidy, 1866), H. dakotensis (Hatcher, 1895), H. oharrai (Jepsen, 1926), H. sicarius (Sinclair & Jepsen, 1927), H. cerebralis (Cope, 1880a), Nimravus brachyops (Cope, 1878), Pogonodon platycopis (Cope, 1879c), P. davisi (Merriam, 1906), Dinictis felina (Leidy, 1854), Dinaelurus crassus (Eaton, 1922), and Nanosmilus kurteni (Martin, 1992). Hoplophoneus dakotensis was partially scored from literature, due to inaccessibility of specimens (Jepsen, 1933; Peigné, 2003). Additionally, the remaining members of the globally described Nimravidae were also included, as determined by Peigné (2003): Nimravus intermedius, Eofelis sp., Dinailurictis bonali, Quercylurus major, Eusmilus bidentatus, and E. villebramarensis. These latter taxa were included to better inform of overall relationships and were scored from the literature (Morea, 1975; Peigné, 2000; Peigné, 2003; Peigné & Brunet, 2001; Peigné & De Bonis, 1999; Ringeade & Michel, 1994).

Figure 19 Mental fossa.

Fossa on the ventral face of the chin. 16(0): no fossa (A); 16(1): fossa present and marked (B). Figured specimens UNSM 1070, Hoplophoneus primaevus (A), YPM 10045, Nimravus brachyops (B).

Figure 20 Incisor shape.

Shape of the lower and upper incisors. 17(0): spatulate incisors, with accessory denticules especially on the lower incisors; I3 slightly caniniform and distinctly larger than the other incisors (A); 17(1): I3, i1–i3 caniniform (B); 17(2): incisors all caniniform; i1 very transversely compressed and i3 nearly as large as the lower canine (C). Figured Specimens USNM 3957, Nimravus brachyops (A), FAM 104823, Dinictis felina (B), UNSM 1072, Hoplophoneus primaevus (C).

Figure 21 C1 length.

Mesial-distal length of C1, measured at the dentine/enamel boundary. 18(0): less than that of P4; 18(1): greater than that of P4. Figured specimen UNSM 1068, Hoplophoneus dakotensis.

Figure 22 Canine serration density.

Serration density of permanent upper canines per millimeter. Measured over an average of 5 mm. 19(0): no serrations; 19(1): 2.0–2.7; 19(2): 2.8 or greater. Figured specimen SDSM 2641, Hoplophoneus primaevus.

Character polarity

Character polarity for this study’s cladistic analyses was determined by the use of multiple real outgroup taxa, since multiple outgroups prevent apomorphic states in the closest sister taxon from being misinterpreted as plesiomorphic states (Nixon & Carpenter, 1993). Three higher taxa (Subfamily or Family) outgroups were chosen containing three exemplar taxa (genera) apiece. Following the recommendations of Prendini (2001): type taxa were included whenever possible (especially when monophyly is contested); when prior cladistic hypotheses were available, exemplars with short branches arising from the root were selected (basal members); and when such cladistic hypotheses were unavailable, exemplars were selected to maximize morphological diversity for the supraspecific taxa they represented. As such, the following taxa were selected as outgroups, based upon the findings of Hunt (1998), Wesley-Hunt & Flynn (2005) and Solé et al. (2014):

∙ Amphicyonidae, Subfamily Daphoeninae: Daphoenus, Daphoenodon, Paradaphoenus

∙ Miacidae: Miacis (any species except M. cognitus, as determined by Wesley-Hunt & Flynn (2005)), Vulpavus,Oodectes

∙ Viverravidae: Viverravus, Didymictis, Protictis.

Optimality criteria

Two optimality criteria were utilized for this study’s analyses, maximum parsimony and posterior probability, to create competing hypotheses of evolutionary history. All determined valid North American taxa, Old World nimravid taxa and outgroups were scored from the aforementioned character list. The resultant character matrix can be seen in Appendix S7.

Figure 23 P3 to P4 height.

Size of P3 vs. size of P4. Measured as a ratio of crown height (base of cingulum to apex of tooth) on adult minimally worn teeth. 20(0): 0.71 and greater; 20(1): 0.5–0.70; 20(2): 0.45 and lower. Figured specimen FAM 104823, Dinictis felina.

Figure 24 Roots of P3.

21(0): P3 two-rooted (A); 21(1): P3 three-rooted (B). The zero state is symplesiomorphic for North American taxa. Figured specimens YPM PU 12558, Dinictis felina (A), MNHN-QU 9477, Quercylurus major (B), modified from Peigné (2003: Fig 10).

Figure 25 Parastyle (ps) on P4.

22(0): absent (A); 22(1): present (B). Figured specimens AMNH 6938 (reversed), Pogonodon platycopis (A), FAM 125660 (reversed), Hoplophoneus primaevus (B).

Figure 26 P4: morphology and size of the protocone (pr) on adult minimally worn teeth.

23(0): protocone well developed with cusp present (A); 23(1): protocone reduced, short, crest-like or absent, resulting from a fusion of the anterior roots (B). Figured specimens AMNH 38805, Dinictis felina (A), USNM 18187 (reversed), Hoplophoneus primaevus (B).

Parsimony.

The parsimony optimality criterion operates under the assumption that the tree with the lowest cost is the most optimal, and forms the basis for the vast majority of phylogenies limited to fossil taxa. This study’s parsimony analysis was conducted with T.N.T. version 1.1 (Goloboff, Farris & Nixon, 2008). Characters were left unordered and the analysis was conducted through an implicit enumeration search.

From the returned most parsimonious trees, a 50% majority-rule consensus tree was constructed. Descriptive statistics and support metrics were then calculated for this consensus tree. Consistency Index (CI) and Retention Index (RI) were calculated in Mesquite version 3.03 (Maddison & Maddison, 2014), while Bremer and jackknife values were calculated in T.N.T. version 1.1 (Goloboff, Farris & Nixon, 2008). For Bremer support all trees with a suboptimal cost up to ten were retained for calculation of the metric, while for jackknife, a traditional search with 1,000 replicates was conducted.

Figure 27 M1: morphology of the tooth and size of the protocone on adult minimally worn teeth.

24(0): M1 triangular in shape, with prominent protocone (pr), paracone (pa) and metacone (mc) (A); 24(1): M1 transversely elongate, with a prominent protocone widely separated from the paracone, and a posterior extension adjacent to centrocrista (cc), forming a reduced “t-shape” (B); 24(2): M1 transversely reduced, crest-like, with low cusps and near absent to absent protocone (C). Figured specimens YPM 10066, Daphoenus vetus (A), UNSM 25512 (reversed), Dinictis felina (B), YPM PU 14999 (reversed), Hoplophoneus primaevus (C).

Figure 28 Lower incisor shape.

Shape of lower incisor arcades. 25(0): lower incisor arcade not or little curved, so as i1 is not visible in lateral view (A); 25(1): lower incisor arcades curved. Figured specimens YPM PU 13587, Dinictis felina (A), YPM PU 11372, Hoplophoneus primaevus (B).

Figure 29 p3: height.

26(0): p3 as tall as or slightly taller than p4 (A); 26(1): p3 lower than p4 (B); 26(2): p3 absent (C). Figured specimens USNM 3957, Nimravus brachyops (A), FAM 125658, Hoplophoneus primaevus (B), YPM PU 11079, Hoplophoneus dakotensis (C).

Posterior probability.

Posterior probability as an optimality criterion for phylogenetic analyses has found far greater use in molecular studies than morphological ones. This is due to the justifiable inclusion of a preferred evolutionary model, such as that of base-pair sequence change. Recently however, studies have started to incorporate fossil taxa along with extant molecular data to produce “tip-dated” time-scaled phylogenies (Arcila et al., 2015; Ronquist et al., 2012; Slater, 2013; Wood et al., 2013). These studies utilized fossil taxa as a means to better inform overall tree topology, but also clarify evolutionary rates through taxa’s first appearances in the stratigraphic record. Other studies have relied solely on morphological data of fossil and extant taxa to produce similar results (Lee et al., 2014; Pyron, 2011). Regardless of the nature of characters utilized, cladistic analysis within a Bayesian framework allows far more evidence to be brought to bear for questions of evolutionary history. Regarding nimravids, no living members of this family exist, so inclusion of molecular data is not possible. However, information regarding stratigraphic first appearances provides a new avenue of inference for the Nimravidae that cannot be readily or justifiably incorporated in a parsimony analysis, and thus finds its first application in this study.

Figure 30 p4: cusps.

27(0): anterior cusp on p4 mesially/distally shorter than the posterior cusp (A); 27(1): anterior cusp more elongated than the posterior cusp (B). Figured specimens YPM PU 11079, Hoplophoneus dakotensis (A), USNM 3957, Nimravus brachyops (B).

Figure 31 p4: height.

28(0): p4 as tall as or taller than the paraconid of m1 (A); 28(1): the main cusp of p4 is lower than the paraconid of m1 (B). Figured specimens USNM 3957, Nimravus brachyops (A), YPM PU 12750 (reversed), Hoplophoneus primaevus (B).

Figure 32 m1: metaconid (mcd).

29(0): present (A); 29(1): absent (B). Figured specimens FAM 125658, Hoplophoneus primaevus (A), AMNH 6938, Pogonodon platycopis (B).

The Bayesian analysis utilizing posterior probability optimality criteria was run in MrBayes version 3.2.5 (Ronquist & Huelsenbeck, 2003). Characters were ordered and set as informative with a gamma parameter utilized for assuming variable rates of evolution across characters. Topology constraints were given to the ingroup and outgroups: Nimravidae, Daphoeninae, Miacidae, and Viverravidae, respectively. A stratigraphic prior was incorporated utilizing the fixed First Appearance Datum (FAD) of each taxon. A fixed calibration was favored over other available options, such as “uniform” or “off-set exponential,” due to the need of setting additional maximum or mean ages of taxon occurrences. The oldest verifiable occurrence of nimravid taxa is for specimens of Hoplophoneus primaevus and Dinictis felina (Bryant, 1996). Based upon all prior cladistic analyses (Bryant, 1996; Peigné, 2003), these taxa are inferred to represent relatively derived members of the Nimravidae and therefore imply that substantial ghost lineages are present for this group. It was therefore viewed as an arbitrary decision if values were to be chosen for other settings beyond a fixed calibration due to the lack of knowledge regarding early nimravid evolution. Utilized FADs and associated citations can be seen in Table 3.

Table 3 FADs and LADs of nimravid taxa.

First Appearance Datum (FAD) and Last Appearance Datum (LAD) of taxa utilized in the phylogenetic analyses of this study. Outgroup data were obtained from the Paleobiology Database, where it was downloaded on 25 November, 2014, using the listed genera names. European taxa stratigraphic ranges were treated as existing in the entire MP zone in which they were found.

Taxa	FAD	Citation	LAD	Citation	
Viverravus	61.7 Ma	Burger (2013)	40.24 Ma	Storer (1984)	
Didymictis	61.7 Ma	Wolberg (1979)	40.4 Ma	Eaton (1985)	
Protictis	66.0 Ma	Schiebout et al. (1987)	40.4 Ma	Flynn & Galiano (1982)	
Miacis	55.8 Ma	Rose et al. (2012)	33.9 Ma	Tabrum, Prothero & Garcia (1996)	
Vulpavus	55.8 Ma	Clyde (1997)	38.0 Ma	Westgate (2001)	
Oodectes	55.8 Ma	Bown et al. (1994)	46.2 Ma	Stucky (1984)	
Daphoenodon	30.8 Ma	Hunt (2002)	15.97 Ma	Hunt (2009)	
Daphoenus	40.4 Ma	Hunt (1998)	20.43 Ma	Hunt (1998)	
Paradaphaenus	33.9 Ma	Hunt (2001)	15.97 Ma	Fremd, Bestland & Retallack (1994)	
Dinaelurus crassus	26.0 Ma	Bryant (1996), Vandenberghe et al. (2012)	23.03 Ma	End of the Oligocene, and Vandenberghe et al. (2012)	
Nimravus brachyops	30.5 Ma	Bryant (1996)	28.7 Ma	Bryant & Fremd (1998)	
Nimravus intermedius	32.63 Ma	Peigné (2003), Vandenberghe et al. (2012)	27.24 Ma	Peigné (2003), Vandenberghe et al. (2012)	
Eofelis	32.63 Ma	Peigné (2003), Vandenberghe et al. (2012)	30.83 Ma	Peigné (2003), Vandenberghe et al. (2012)	
Dinailurictis bonali	30.83 Ma	Peigné (2003), Vandenberghe et al. (2012)	27.24 Ma	Peigné (2003), Vandenberghe et al. (2012)	
Quercylurus major	28.82 Ma	Peigné (2003), Vandenberghe et al. (2012)	27.24 Ma	Peigné (2003), Vandenberghe et al. (2012)	
Dinictis felina	35.5 Ma	Bryant (1996)	23.03 Ma	End of the Oligocene, Bryant (1996)Vandenberghe et al. (2012)	
Pogonodon platycopis	32.1 Ma	Start of Whitneyan, Vandenberghe et al. (2012)	28.7 Ma	Bryant & Fremd (1998)	
Pogonodon davisi	32.1 Ma	Bryant (1996)Vandenberghe et al. (2012) start of Whitneyan	27.5 Ma	Bryant & Fremd (1998)	
Nanosmilus kurteni	33.89 Ma	Bryant (1996)Vandenberghe et al. (2012) start of Orellan	32.1 Ma	Bryant (1996)Vandenberghe et al. (2012) end of Orellan	
Hoplophoneus oharrai	34.0 Ma	Bryant (1996), Vandenberghe et al. (2012)	33.89 Ma	Bryant (1996), Vandenberghe et al. (2012)	
Hoplophoneus primaevus	35.5 Ma	FAD of H. mentalis, Bryant (1996)	30.25 Ma	Bryant (1996)	
Hoplophoneus occidentalis	33.25 Ma	Bryant (1996)	30.5 Ma	Bryant (1996)	
Hoplophoneus dakotensis	30.5 Ma	Bryant (1996)	29.75 Ma	Jepsen (1933), Vandenberghe et al. (2012)	
Hoplophoneus sicarius	33.89 Ma	Bryant (1996), Vandenberghe et al. (2012)	32.1 Ma	Bryant (1996), Vandenberghe et al. (2012)	
Hoplophoneus cerebralis	34.0 Ma	Accompanying age description of YPM PU 16271, Vandenberghe et al. (2012)	28.0 Ma	Bryant (1996), Vandenberghe et al. (2012)	
Eusmilus bidentatus	33.77 Ma	Peigné (2003), Vandenberghe et al. (2012)	32.63 Ma	Peigné (2003), Vandenberghe et al. (2012)	
Eusmilus villebramarensis	32.63 Ma	Peigné (2003), Vandenberghe et al. (2012)	30.83 Ma	Peigné (2003), Vandenberghe et al. (2012)	

For modeling rate variation across branches, an uncorrelated (igr) relaxed clock was used. The root age was given a uniform constraint from the first viverravid appearance (minimum age), to the base of Aquilan (the first North American Land Mammal Age). This was considered a generous span of time to incorporate the “true” first appearance of all clades utilized in this study. The resultant command file with parameters utilized in this study can be seen in Appendix S8.

Figure 33 m1: trigonid (trgd) proportion of m1.

30(0): trigonid length is ≤70% of total length; 30(1): 77–87%; 30(2): 88% and higher. Figured specimen UNSM 2509-59, Pogonodon platycopis.

Figure 34 Cheek tooth serrations.

Serrations on adult minimally worn cheek teeth. 31(0): absent (A); 31(1): present (B). Figured specimens AMNH 6938, Pogonodon platycopis (A), FAM 125658, Hoplophoneus primaevus (B).

The analysis used four replicate runs of five million generations, with sampling every 1,000 generations; each run consisted of one unheated and three incrementally heated chains. A burn-in of 25% was utilized, and the analysis was suspended at the end of five million generations where the average standard deviation of split frequencies reached 0.006635. This is below the 0.01 threshold stated as a level of “very good” convergence by the authors of the program (Ronquist & Huelsenbeck, 2003). From the results of this analysis a 50% majority-rule consensus tree, generated from clade posterior values within the stationary pool, was constructed.

Results

Specific validity

Dinictis

The original linear morphometric dataset for this genus presented 29.31% missing data. Subsequent to imputation, the previously described cluster analysis was performed (Fig. 37). From this dendrogram two differential DFAs were conducted utilizing group assignment as determined by the figured node-associated values. The resultant classification accuracy for the test set of the two-fold subdivision was 100% (Fig. 38), while only 64.29% for the four-fold subdivision. However, even though 100% discrimination was returned for the two-fold subdivision of this genus, no differential character state combinations were found between either morphogroup. Within the light of the chosen species concept (PSC), this suggests that only intraspecific variation for a single species is being seen in the performed morphometric analyses.

Figure 35 Ratio of tibia to femur, measured as length of elements at most distal articular surface, Tibia/Femur X 100.

32(0): 87% and higher; 32(1): 83% and lower. Figured specimen UNSM 1888-38, Hoplophoneus sp.

Pogonodon

The original linear dataset for this genus consisted of 25% missing data. Following imputation the resultant cluster analysis of the Pogonodon genus can be seen in Fig. 39. From this dendrogram both two and three-fold subdivisions were tested. Classification accuracy for two groups was determined to be 90% (Fig. 40), while only 80% for three groups. Within the two-fold subdivision, thirty-three characters (and their associated states) were found to be universally shared within each morphogroup, yet differential between the two, implying the presence of two contained species for this genus.

Figure 36 Tarsus articulation.

Articulation between the calcaneum (cal) and navicular (nav). 33(0): absent (A); 33(1): present (B). Figured specimens AMNH 38981, Hoplophoneus primaevus (A), FAM 62151, Nimravus brachyops (B). Astragalus (ast).

Figure 37 Dinictis cluster analysis.

Cluster analysis of the Dinictis genus based upon ten cranial and dentary measurement variables. Discriminant function analyses were performed utilizing the group assignments depicted (two and four fold subdivisions, the latter indicated by parenthetical numbers). Asterisks equate to holotype specimens.

Nimravus

The Nimravus genus presented 15.83% missing data in the original morphometric dataset. After imputation the cluster analysis seen in Fig. 41 was produced. The figured two-fold subdivision was tested via DFA and returned only 33.33% classification accuracy for these morphogroups (Fig. 42). This, combined with no differential character states between these morphogroups implies the existence of only one North American species for this genus.

Hoplophoneus

The Hoplophoneus morphometric dataset presented the most missing data of any genus (41.18%), though additionally contained the most specimens at eighty-five. The holotype specimens of H. dakotensis and H. marshi were removed prior to imputation, as was UNSM 1068 (historically diagnosed to H. dakotensis). These specimens exhibited substantial degrees of missing data (or problematic values in the case of H. marshi due to the specimen being juvenile) which created difficulties in subsequent imputation, specifically, imputation of a cranium from a dentary, or vice-versa, for the H. dakotensis specimens.

Figure 38 Dinictis DFA.

Discriminant Function Analysis (DFA) of the Dinictis genus utilizing group assignment based upon two fold subdivision seen in Fig. 36. Group assignment classification accuracy of the test set was 100%.

Figure 39 Pogonodon cluster analysis.

Cluster analysis of the Pogonodon genus based upon four dentary measurement variables. Discriminant function analyses were performed utilizing the group assignments depicted (two and three fold subdivisions, the latter indicated by parenthetical numbers). Asterisks equate to holotype specimens.

Following imputation, cluster analysis was performed (Fig. 43). However, prior to the differential DFA analyses the holotype specimens of H. sicarius and H. oharrai were removed due to being determined as unique (see ‘Discussion’), and the inability to perform a DFA on group sizes of one. Three differential sets of DFAs were performed utilizing group assignment as determined from the cluster analysis, specifically, three, four and six-fold subdivisions were tested. Classification accuracy for these analyses was 97.56% (Fig. 44), 95.12% and 82.50% respectively. While the four group DFA produced highly supported results, no differential character states between morphogroup three or four were discovered, the same with the six group analysis and morphogroups three through six. This implies that like the results of the Dinictis DFA, only intraspecific variation for a single species (morphogroup three of the three-fold analysis) is being seen and not distinct species.

Cladistic analyses

Parsimony

The parsimony analysis returned 222 most parsimonious trees with a cost of 75, C.I. = 0.55, R.I. = 0.82, (Fig. 45). Two major clades were retrieved for this phylogenetic hypothesis, the Nimravini and Hoplophoneini, located at the base of the Dinaelurus crassus through N. intermedius clade, and Nanosmilus kurteni through Eusmilus villebramarensis clade respectively, both of which display an overall pectinate arrangement. The name of these clades is derived from similar relationships retrieved in Bryant (1996), and his application of accompanying tribal designation (see Fig. 1). The European taxa of Eofelis, Dinailurictis, and Quercylurus major were also retrieved as a monophyletic group, as were the species of Pogonodon platycopis, and P. davisi. Substantial support (Bremer and jackknife values of 5 and 98% respectively) was recovered for the Hoplophoneini clade, with significantly smaller values for the rest of the tree topology.

Figure 40 Pogonodon DFA.

DFA histogram generated from the most-inclusive group assignment of the cluster analysis seen in Fig. 38. Group assignment accuracy based upon jackknife cross-validation for the above DFA was 90%.

Figure 41 Nimravus cluster analysis.

Cluster Analysis of specimens belonging to the Nimravus genus. Discriminant Function Analysis was performed on the two-fold subdivision depicted by node associated values.

Posterior probability

The 50% majority-rule consensus tree, generated from clade posterior values within the stationary pool, returned less overall resolution for the ingroup than that of the parsimony analysis (Fig. 46). However, similar patterns were found in the recovery of the already described Nimravidae tribes, the Nimravini, and Hoplophoneini, with both displaying 100% support. Relationships within the Nimravini were identical to that of the parsimony analysis. Conversely, relationships within the Hoplophoneini differed somewhat. Hoplophoneus primaevus, and H. occidentalis were recovered as a sister-group, as were H. sicarius, and H. cerebralis. Additionally, the species of Hoplophoneus sicarius through Eusmilus villebramarensis were also determined to form a clade, though interrelationships of this group could not be thoroughly resolved. Finally, Nanosmilus kurteni, Hoplophoneus oharrai and H. dakotensis were unable to be resolved within the Hoplophoneini, comprising a polytomic arrangement at the base of the clade. The European taxa Eofelis, Dinailurictis bonali, and Quercylurus major were recovered as a well-supported clade (95%), as were the species of Pogonodon platycopis, and P. davisi (79%). Combined with Dinictis felina, these taxa could not be resolved with respect to each other, though appear intermediate between the aforementioned tribes.

Discussion

With this study’s implementation of the Phylogenetic Species Concept (Nixon & Wheeler, 1990), species diagnoses are derived from the unique character state combination presented in the data matrix of the cladistic analyses, and aforementioned character description of the character list. However, to better facilitate identification of and finer-scale differences between species, autapomorphies, and descriptive statistics are also presented when possible of sub-adult to adult specimens, the basis for this study. Generic reassignment was quite common in the early taxonomic history of most nimravid species, and the structure of the presented systematic paleontology sections gives a comprehensive list of all generic referrals for all North American species. Current non-valid generic referrals are denoted within quotations.

Figure 42 Nimravus DFA.

DFA histogram of Nimravus specimens generated from the group assignment of the cluster analysis seen in Fig. 40. Although there appears to be a high degree of group discrimination in the generated histogram of the training set, when predictions from this model were applied to the test set, group assignment accuracy was only 33.33%.

Systematic Paleontology

MAMMALIA (Linnaeus, 1758)	
CARNIVORA (Bowdich, 1821)	
NIMRAVIDAE (Cope, 1880b)	

Genus Dinictis (Leidy, 1854)

Referred taxa

Dinictis (Leidy, 1854) Daptophilus (Cope, 1873b)	

Type and only referred species —Dinictis felina (Leidy, 1854)

Distribution —Middle to Late Chadronian (Ch2-3) of Montana (Renova Fm.), Nebraska (Chadron Fm.), Saskatchewan (Cypress Hills Fm.), South Dakota (Chadron Fm.), Wyoming (White River Fm., Chadron Mbr.); Orellan of Colorado (Brule Fm., Orella Mbr.), Nebraska (Brule Fm., Orella Mbr.), North Dakota (Brule Fm.), South Dakota (Brule Fm., Scenic Mbr.), Wyoming (White River Fm., Brule Mbr.); Whitneyan of Nebraska (Brule Fm., Whitney Mbr.), Saskatchewan (Cypress Hills Fm.), South Dakota (Brule Fm., Poleslide Mbr.); early Arikareean (Ar1-2) of Oregon (John Day Fm., Turtle Cove Mbr.).

Figure 43 Hoplophoneus cluster analysis.

Cluster analysis of specimens belonging to the genus Hoplophoneus. Discriminant function analyses were performed utilizing the group assignments depicted (three, four and six-fold subdivisions, indicated by superscripts). Asterisks equate to holotype specimens, while those in red are specimens that have been traditionally diagnosed to H. mentalis.

Diagnosis —Sutural contact between the lacrimal and jugal; absence of lateral and medial fossae on the zygomata; broadly circular zygomata in dorsal view; presence of discrete petrobasilar and posterior lacerate foramina; reduced mastoid with large plate-like paroccipital process; posterior lip of the glenoid socket projects more ventrally than anterior lip; oblique angle between the braincase and axial plane of the cranium; posteriorly orientated posterior border of coronoid process; short genial flange, between 22 and 31% of total dentary length; absence of fossa on ventral face of chin; I3, i1–i3 caniniform; mesial-distal length of C1 less than that of P4; ratio of height of P3–P4, 0.71 and greater; parastyle absent from P4; P4 protocone well developed with cusp present; M1 transversely elongate, with a prominent protocone widely separated from the paracone, and a posterior extension adjacent to centrocrista, forming a reduced “t-shape”; p3 height is as tall or slightly taller than p4; anterior cusp on p4 mesially/distally shorter than the posterior cusp; p4 as tall as or taller than the paraconid of m1; m1 metaconid present; trigonid proportion of m1 is 77–87%; serrations present on adult minimally worn cheek teeth; ratio of tibia to femur 87% and higher; articulation between the calcaneum and navicular absent.

Figure 44 Hoplophoneus DFA.

Discriminant function analysis for the three groups depicted in Fig. 42. Resultant DFA group-assignment accuracy of the test set for these three groups was 97.56%. Cluster one equates to specimens diagnosed to H. occidentalis, cluster two H. cerebralis, and cluster three H. primaevus.

Discussion—Leidy (1869) viewed Dinictis as a lesser-derived form of the more saber-toothed Hoplophoneus (=Drepanodon), but with dentition more analogous to certain mustelines. Cope (1883) supported the notion of Dinictis as a primitive saber-tooth, and the most generalized of the nimravids; however, he rejected the musteline affinities suggested by Leidy. Scott & Jepsen (1936) cited resemblance to early “dogs,” such as Daphoenus, with the greater number of teeth, including a robust upper molar, presence of a second lower molar, and “dog-like” sectorial dentition, when compared to members of the Felidae. Scott and Jepsen referred seven species to Dinictis, but stated that the taxonomy of these species was in confusion requiring additional material to unravel. Bryant (1996) outlined a cursory revision of the genus, based upon the validity of characters utilized in species diagnoses, resulting in two valid taxa. Cluster analysis of morphometric data for this genus (Fig. 37) found significant support (100% discrimination) for two groups (Fig. 38). However, no morphological characters and their respective states were found to align themselves with this distinction. Through the utilization of the PSC, this implies that discrimination of the DFA was based upon intraspecific variation of a single species for this genus, such as sexual dimorphism, temporal or regional variation.

Dinictis felina (Leidy, 1854)

Referred taxa

Dinictis felina (Leidy, 1854)	
Daptophilus squalidens (Cope, 1873b)	
Dinictis squalidens (Cope, 1879b)	
Dinictis cyclops (Cope, 1879b)	
Dinictis fortis (Adams, 1895)	
Dinictis bombifrons (Adams, 1895)	
Dinictis paucidens (Riggs, 1896)	

Type —AMNH 455, Partial cranium and dentary.

Referred specimens —Numerous, see Appendix S3.

Distribution —Same as genus.

Diagnosis —Same as genus.

Autapomorphic and descriptive features —A nimravid of moderate size, with a basilar length between 112 and 182 mm, mean of 141, (n = 30); lambdoid crest angle ranging from 130 to 148 degrees, mean of 138, (n = 27); serration density per millimeter ranging from 2.8 to 4.8, mean of 3.6, (n = 22); C1 compression ranging from 1.46 to 2.44, mean of 1.87, (n = 24); variable presence of lacrimal processes; absent anterior cusp of P3.

Discussion —Leidy (1854) based Dinictis felina on a partial cranium and dentary from the White River Group of the Dakota Territory (modern day South Dakota). Leidy (1869) noted overall resemblance of this species to that of Hoplophoneus (=Drepanodon) primaevus. Cope (1873a) established Daptophilus squalidens on a fragmentary upper deciduous canine, and dentary with m1 and deciduous premolars. Even with noted similarity to that of Dinictis, a new genus was erected to contain the species due to the apparent absence of the m2. Subsequently, Cope (1879b) and Cope (1880b) rectified the generic assignment of this species to Dinictis with discovery of evidence for this tooth’s existence. Cope (1883) diagnosed D. squalidens on a single-rooted p2, and shorter anterior–posterior length of the dentary tooth-row. Sinclair (1924) differentiated this species from D. felina primarily on size. Scott & Jepsen (1936) stated that this species is the smallest in the genus, but noted variability on that score and therefore utilized relative cranial features to distinguish it. Dinictis squalidens was said to possess a “…horizontal brain-case…[and] almost absence, of the glenoid and mastoid pedicles” (p. 123). D. cyclops was erected on a mostly complete cranium and dentary exhibiting subtle differences in many aspects of its anatomy (Cope, 1879b). Of particular note to Cope was the number of roots of the upper and lower P/p2s, one and two respectively, the compression of the upper canine, and the shortness of the rostrum. Toohey (1959) noted the similarity of USNM 16558, identified as D. cyclops by Hough (1953), to Nimravus brachyops. Confusingly, he synonymized the species in his revision of the Nimravus genus. Bryant (1996) cited a reduced p3 when compared to D. felina. Adams (1895) erected two new species, D. fortis and D. bombifrons, but subsequently synonymized them into one, D. fortis (Adams, 1896b). These species were characterized by their more robust size, shorter rostrum, lack of anterior cusp to the p3, and larger upper canines, when compared to D. felina. Riggs (1896) established D. paucidens on a partial skeleton lacking m2, shape of the P4, and supposed presence of only two incisors in the dentary.

The type of D. fortis was unable to be examined during the course of this study, but the type of D. bombifons was, and being synonymized to D. fortis by Adams himself, should have afforded an adequate surrogate for these large specimens. With the inclusion of D. cyclops to this species, the range extension of D. felina encompasses a possible time of 12.5 Ma. However, the exact temporal and stratigraphic occurrence of the type remains uncertain (Bryant & Fremd, 1998).

Genus Pogonodon (Cope, 1880b)

Referred taxa

“Hoplophoneus” (sensu Cope, 1879c)	
Pogonodon (Cope, 1880b)	
“Dinictis” (sensu Adams, 1896b)	
	

Type species —Hoplophoneus platycopis (Cope, 1879c)

Referred species —Pogonodon davisi (Merriam, 1906)

Distribution —Oligocene of Wyoming (White River Fm.); Orellan of South Dakota (Brule Fm., Scenic Mbr.); Whitneyan of Nebraska (Brule Fm., Whitney Mbr.), South Dakota (Brule Fm., Poleslide Mbr.); early Arikareean (Ar1-2) of Oregon (John Day Fm., Turtle Cove and Kimberly Mbrs.).

Diagnosis —Sutural contact between the lacrimal and jugal; absence of lateral and medial fossae on the zygomata; zygomata triangular in dorsal view; reduced mastoid with large plate-like paroccipital process; anterior lip of the glenoid socket absent; oblique angle between the braincase, sans sagittal crest, and axial plane of the cranium; posteriorly orientated posterior border of coronoid process; short genial flange, between 22 and 31% of total dentary length; absence of fossa on ventral face of chin; I3, i1–i3 caniniform; serration density of permanent upper canines per millimeter 2.0–2.7; ratio of height of P3–P4, 0.71 and greater; parastyle absent from P4; P4 protocone well developed with cusp present; M1 transversely elongate, with a prominent protocone widely separated from the paracone, and a posterior extension adjacent to centrocrista, forming a reduced “t-shape”; p3 height is as tall or slightly taller than p4; anterior cusp on p4 mesially/distally shorter than the posterior cusp; p4 as tall as or taller than the paraconid of m1; m1 metaconid absent; trigonid proportion of m1 is 77–87%; serrations present on adult minimally worn cheek teeth.

Discussion —Cope (1880b) viewed Pogonodon as an intermediate form between Dinictis and Hoplophoneus, specifically noting the lack of m2 and possession of p2, whereby Dinictis (typically) possesses both of these structures and Hoplophoneus lacks them. Adams (1896b) held Pogonodon as a synonym of Dinictis, noting similarity of the tooth structure, and variability of the possession of m2, including polymorphism for a specimen of Dinictis. Subsequent authors (e.g., Merriam, 1906; Matthew, 1910; Thorpe, 1920; Eaton, 1922) held similar views to Cope with the uniqueness of this taxon, and maintained that it should at least be viewed as a valid subgenus. Bryant (1996) offered a preliminary revision of the taxon, upholding the genus and citing one valid species, though left open the possibility of additional species, specifically Dinictis eileenae (Macdonald, 1970).

The paucity of specimens for this taxon made the standard set of morphometric analyses utilized by this study difficult. A particular challenge was the incorporation of two type specimens represented by isolated dentaries (P. cismontanus and P. eileenae). To circumvent these problems, the cluster analysis (Fig. 39) and associated DFA (Fig. 40) were performed on a reduced dataset of four mandibular variables. Significant support (90% DFA discrimination) was found for a two-fold subdivision within this genus. This subdivision was further aligned with shared differences in possession of character states, representing justification for two valid species for this genus.

Pogonodon platycopis (Cope, 1879c)

Referred taxa

“Hoplophoneus” platycopis (Cope, 1879c)	
Pogonodon platycopis (Cope, 1880a)	
“Dinictis” platycopis (Adams, 1896b)	
Pogonodon cismontanus (Thorpe, 1920)	
“Dinictis” cismontanus (Scott & Jepsen, 1936)	
	

Type —AMNH 6938, cranium and dentary.

Referred specimens —YPM 10053, UNSM 2509-59.

Distribution —Oligocene of South Dakota (White River Grp.), Whitneyan of Nebraska (Brule Fm., Whitney Mbr.); early Arikareean (Ar1) of Oregon (John Day Fm., Turtle Cove Mbr.).

Diagnosis —Characters of Pogonodon plus: basilar length 205 mm and greater, mesial-distal length of upper canine greater than that of P4.

Autapomorphic and descriptive features —A nimravid of large size, with a basilar length of 215 mm, (n = 1). Extremely large lambdoid crest angle of 164 degrees, (n = 1); serration density per millimeter of upper canine 2.2, (n = 1); C1 compression of 1.74, (n = 1).

Discussion —Cope (1879c) established Hoplophoneus platycopis on a large, rather well preserved, cranium and dentary originating from the John Day Formation of Oregon. The generic assignment of Hoplophoneus was likely due to the lack of m2s in the specimen, and the specific diagnosis was put in terms of the prominent anterior premolars. Thorpe (1920) based Pogonodon cismontanus on a complete left dentary from the Big Badlands of South Dakota, stating “The size is close to that of the type of the genus, P. platycopis Cope, although in nearly every dimension this specimen is slightly smaller” (p. 223). However, no additional diagnosing features were given.

Pogonodon cismontanus possesses the same preserved character states of any member of the Pogonodon genus. However, cluster and discriminant function analysis of preserved dentary variables finds it comparable to that of the type of P. platycopis. Therefore, they are placed in synonymy.

The stratigraphic range of this species appears to cover approximately 3.4 Ma, whereby UNSM 2509-59 comes from the Whitney Member of the Nebraska Brule Formation, and the type specimen occurring no earlier than 28.70 Ma (Bryant & Fremd, 1998). However, the type of Pogonodon cismontanus is described as simply coming from the White River Group. With such poor constraint, and lack of additional specimens, future work is needed to better approximate the true range of this taxon.

Pogonodon davisi (Merriam, 1906)

Referred taxa

Pogonodon davisi (Merriam, 1906)	
“Hoplophoneus” davisi (Matthew, 1910)	
Pogonodon serrulidens (Eaton, 1922)	
Dinictis eileenae (Macdonald, 1970)	

Type —CMP 789, cranium.

Referred specimens —SDSM 2865, AMNH 62124, FAM 102155, FAM 62026, FAM 62019, FAM 62042, AMNH 1403, LUSK 309-2729, YPM 10520, LACM 9195.

Distribution —Oligocene of Wyoming (White River Fm.); Whitneyan of Nebraska (Brule Fm., Whitney Mbr.), South Dakota (Brule Fm. Poleslide Mbr.); early Arikareean (Ar1) of South Dakota (Sharps Fm.) and (Ar1-2) of Oregon (John Day Fm., Turtle Cove and Kimberly Mbrs.).

Diagnosis —Characters of Pogonodon plus: basilar length 110–200 mm; mesial-distal length of canine less than P4; ratio of tibia to femur 83% and lower; articulation between the calcaneum and navicular present.

Autapomorphic and descriptive features —A nimravid of moderate size, with a basilar length between 169 and 198 mm, mean of 186, (n = 7); lambdoid crest angle ranging from 134 to 148 degrees, mean of 144, (n = 4); serration density per millimeter of upper canine 2.7, (n = 2); C1 compression ranging from 1.51 to 1.64, mean of 1.58, (n = 2); variable presence of lacrimal processes.

Discussion —Merriam (1906) established Pogonodon davisi on a cranium from the upper John Day Formation. Diagnosis for this species was derived from the smaller overall size, shape of the M1, and proportion of the posterior portion of the cranium, compared to those of P. platycopis. Matthew (1910) chose to place P. davisi within Hoplophoneus based upon the published figures of Merriam (1906). Eaton (1922) named Pogonodon serrulidens on a sub-adult cranium missing most of the rostrum, fragmentary right dentary, and associated postcranial elements. Eaton saw close affinity of his species to that of P. davisi, but noted differential characteristics in the form of smaller size, lower sagittal crest, and greater compression of the upper canines. Macdonald (1970) established Dinictis eileenae on a fragmentary right dentary from the Sharps Formation of South Dakota, listing a long post canine diastema, single rooted p2, and well developed p3 as diagnosable features. Bryant (1996) chose to synonymize P. davisi, and all the other Pogonodon taxa, into a single species, P. platycopis, in his cursory revision of the genus.

This study found validity in a second smaller Pogonodon species, based upon the returned results of the cluster and discriminant function analysis on dentary associated variables and subsequent character analysis, specifically the mesial-distal length of the upper canine compared to that of the P4, and basilar length. UPGMA cluster analysis grouped the examined cast of the type specimen of Dinictis eileenae (USNM 25146) along with other specimens associated with this taxon. However, it can be seen from the raw measurement values that the chosen imputation technique for this study may have been primarily responsible for this outcome, for the measurements for the preserved m1 are most similar to that of the type of P. platycopis. Therefore, some hesitancy should be given for the definitive assignment of Dinictis eileenae to this species.

For this smaller species it would seem most fitting to resurrect the name of Pogonodon davisi due to its precedence, and original diagnosis noting smaller size and “P4…not exceeding the superior canine in anterior–posterior diameter” (Merriam, 1906, p.53). P. serrerrulidens was unable to be examined directly, but given the published dimensions and description, very likely belongs to this taxon, and therefore is placed in synonymy.

HOPLOPHONEINI Kretzoi (1929)

Definition —The most recent common ancestor of Nanosmilus kurteni, Hoplophoneus oharrai, H. primaevus, H. occidentalis, H. sicarius, H. dakotensis, H. cerebralis, “Eusmilus” bidentatus, “E.” villebramarensis and all of its descendants.

Included Genera —Hoplophoneus, Nanosmilus.

Diagnosis —Synapomorphies: petrobasilar and posterior lacerate foramina form two distinct grooves; large tabular mastoid with reduced to near absent paroccipital process; incisors all caniniform, i1 very transversely compressed and i3 nearly as large as the lower canine; parastyle present on the P4; lower incisor arcade curved.

Discussion —Both Bryant (1996) and Peigné (2003) recovered substantial support for clades consisting of the derived saber-tooth nimravid taxa, though the former did not incorporate Nanosmilus as one of its operational taxonomic units. The cladistic analyses of this study also returned strong support for this clade (Bremer support and jackknife values of 5 and 98, respectively, for parsimony analyses, and 100% consensus of posterior probability for the Bayesian analysis). Bryant (1996) referred to this clade as the Hoplophoneini, a tribal designation further utilized by Peigné (2003). It would therefore be fitting to retain this group’s label to avoid confusion.

Genus Hoplophoneus (Cope, 1874)

Referred taxa

“Machairodus” (sensuLeidy, 1851)	
“Drepanodon” (sensuLeidy, 1857)	
Hoplophoneus (Cope, 1874)	
Dinotomius (Williston, 1895)	
“Eusmilus” (sensuHatcher, 1895)	
“Eusmilus” (sensuSinclair & Jepsen, 1927)	
“Drepanodon” (sensuScott & Jepsen, 1936)	
“Eusmilus” (sensuToohey, 1959)	
Ekgmoiteptecela (Macdonald, 1963)	
“Eusmilus” (sensuMorea, 1975)	
“Eusmilus” (sensuPeigné, 2003)	

Type species —Hoplophoneus oreodontis (Cope, 1874)

Included species —Hoplophoneus oharrai, H. primaevus, H. occidentalis, H. sicarius, H. dakotensis, H. cerebralis.

Distribution —Middle to late Chadronian (Ch2-3) of Nebraska (Chadron Fm.), South Dakota (Chadron Fm., Crazy Johnson and Peanut Peak Mbrs.), Wyoming (White River Fm., Chadron Mbr.), ?Saskatchewan (Cypress Hills Fm.); Orellan of Colorado (Brule Fm., Orella Mbr.), Nebraska (Brule Fm., Orella Mbr.), North Dakota (Brule Fm.), South Dakota (Brule Fm., Scenic Mbr.), Wyoming (White River Fm., Brule Mbr.); Whitneyan of Nebraska (Brule Fm., Whitney Mbr.), South Dakota (Brule Fm., Poleslide Mbr.); early Arikareean (Ar1-2) of California (Sespe Fm.), Oregon (John Day Fm., Turtle Cove Mbr.), South Dakota (Sharps Fm.), and Wyoming (White River Fm.).

Diagnosis —Absence of lateral and medial fossae on the zygomata; triangular zygomata in dorsal view; petrobasilar and posterior lacerate foramina form two distinct grooves; large tabular mastoid with reduced to near absent paroccipital process; incisors all caniniform, i1 very transversely compressed and i3 nearly as large as the lower canine; serration density of permanent upper canines per millimeter, 2.8 or greater; parastyle on P4; P4 protocone reduced, short, crest-like or absent; M1 transversely reduced, crest-like, with low cusps and near absent to absent protocone; lower incisor arcade curved; p3 height is as tall or slightly taller than p4; anterior cusp on p4 mesially-distally shorter than the posterior cusp; the maincusp of the p4 is lower than the paraconid of m1; m1 metaconid present; serrations present on adult minimally worn cheek teeth; ratio of tibia to femur 83% and lower; articulation between the calcaneum and navicular absent.

Discussion —Cope (1883) diagnosed Hoplophoneus primarily on pronounced saber-tooth adaptations, and the dental formula, whereby this taxon lacked the p2 and m2, features historically utilized in distinguishing Dinictis and Pogonodon from Hoplophoneus. Subsequent to this, the number of contained species rose from four to five (depending on author) to fourteen, when Scott & Jepsen (1936) offered the first tentative revision. Although they synonymized H. robustus and H. insolens to Drepanodon (=Hoplophoneus) occidentalis, they still voiced concerns over the assuredly too large number of contained species. Simpson (1941) performed the first major revision of the genus through use of comparing metrical data, and qualitative features. This reduced the total number of valid species to a tentative four. Hough (1949) followed on the heels of Simpson with an expanded study (nineteen newly discovered specimens) employing the same techniques. Hough largely followed Simpson’s conclusions, but argued for two subspecies of H. primaevus based upon differential cranial indices and associated geographic distribution. Morea (1975) defined new criteria in distinguishing the Hoplophoneus and Eusmilus genera, something which proved quite difficult to prior revisionists (e.g., Jepsen, 1933). Morea argued for the importance of the relative angle between the dorsal surfaces of the face and braincase of the cranium, sans sagittal crest, a departure from the primary dental characteristics previously used. With this feature he rectified the species of Eusmilus dakotensis (Hatcher, 1895), and Eusmilus sicarius (Sinclair & Jepsen, 1927), to Hoplophoneus.

The returned phylogenies of this study follow, in large part, that of the published phylogenies of Bryant (1996), and Peigné (2003), in that a Hoplophoneini clade is returned containing the combined Hoplophoneus and Eusmilus taxa in a pectinate arrangement. This relationship renders Hoplophoneus paraphyletic. From these arrangements there are only two ways of rendering monophyletic genera within the Hoplophoneini: (1) maintain the validity of Eusmilus and grant every other species previously allotted to Hoplophoneus a differential generic assignment, or (2) synonymize both Hoplophoneus and Eusmilus. The simplest resolution to this problem would be the latter option. As such, due to precedence, (Hoplophoneus Cope, 1874; Eusmilus Gervais, 1876) Hoplophoneus will become the valid generic designation for these taxa. The scope of this study was limited to North American representatives of the Nimravidae, and thus conclusions drawn on the validity of the Eusmilus genus for European members must be postponed pending further analysis and potentially discernable distinctions.

Hoplophoneus primaevus (Leidy, 1851)

Referred taxa

“Machairodus” primaevus (Leidy, 1851)	
“Drepanodon” primaevus (Leidy, 1857)	
“Machaerodus” oreodontis (Cope, 1873a)	
Hoplophoneus oreodontis (Cope, 1874)	
Hoplophoneus primaevus (Cope, 1879b)	
Hoplophoneus robustus (Adams, 1896a)	
Hoplophoneus insolens (Adams, 1896a)	
Hoplophoneus marshi (Thorpe, 1920)	
Hoplophoneus latidens (Thorpe, 1920)	
Hoplophoneus molossus (Thorpe, 1920)	
Hoplophoneus mentalis (Sinclair, 1921)	
Hoplophoneus primaevus primaevus (Hough, 1949)	
Hoplophoneus primaevus latidens (Hough, 1949)	

Type —USNM 99, cranium and partial dentary.

Referred specimens —Numerous, see Appendix S3.

Distribution —Middle to late Chadronian (Ch2-3) of Nebraska (Chadron Fm.), South Dakota (Chadron Fm., Peanut Peak Mbr.), Wyoming (White River Fm., Chadron Mbr.), ?Saskatchewan (Cypress Hills Fm.); Orellan of Colorado (Brule Fm., Orella Mbr.), Nebraska (Brule Fm., Orella Mbr.), North Dakota (Brule Fm.), South Dakota (Brule Fm., Scenic Mbr.), Wyoming (White River Fm., Brule Mbr.); Whitneyan of Nebraska (Brule Fm., Whitney Mbr.), South Dakota (Brule Fm., Poleslide Mbr.).

Figure 45 Parsimony cladistic analysis.

Majority-rule consensus of 222 trees, cost 75, unordered characters. Consensus values are located below nodes, while above are Bremer support (in red), and jackknife values ≥50% .

Diagnosis —Characters of Hoplophoneus plus: sutural contact absent between the lacrimal and jugal; medial ridge of palate present; postglenoid foramen present; posterior lip of the glenoid socket projects more ventrally than the anterior lip; oblique angle between the braincase, disregarding the sagittal crest, and axial plane of the cranium; anterior dentary position elevate above cheek tooth row; vertically orientated posterior border of coronoid process; deep genial flange between 32 and 50% of total dentary length; mesial-distal length of C1 less than that of P4; ratio of height between P3 and P4 0.50 to 0.70; p3 lower than p4; trigonid proportion of m1, 88% or greater.

Autapomorphic and descriptive features —A nimravid of moderate size with basilar length between 121 and 189 mm, mean of 154, (n = 75); lambdoid crest angle ranging from 96 to 133 degrees, mean of 116, (n = 76); serration density per millimeter ranging from 2.9 to 6.2, mean of 4.4, (n = 40); C1 compression ranging from 1.55 to 2.35, mean of 1.87, (n = 44); lacrimal process present; variable anterior cusp of P3.

Discussion —Leidy (1851) described the first nimravid specimen and assigned it to the Felidae taxon Machairodus, under M. primaevus. It was not until Cope (1880b) that it was realized this specimen (and additional species) belonged to a distinct family. However, by this point in time Cope had already rectified the type specimen to a new genus, Hoplophoneus, based upon a more primitive m1 than that of Machairodus (Cope, 1879b, expanded upon in Cope, 1883). Simpson (1941), determined that specimens referred to Hoplophoneus oreodontis-primaevus and H. robustus-insolens formed two distinct groups, but viewed these groups as sexual dimorphism of the same species (female and male respectively). H. latidens and H. marshi were also synonymized with this former series, but H. molossus was judged valid for displaying osteology very robust for its diminutive size. Hough (1949) synonymized H. molossus with H. primaevus, but split the latter into two subspecies: H. primaevus primaevus (from South Dakota), and H. primaevus latidens (from Nebraska, Wyoming, and Colorado). Morea (1975) maintained the findings of Hough (1949) in his study, but rejected the validity of her subspecies.

This study arrived at the same general conclusions of previous authors concerning H. primaevus, but differs in the inclusion of H. mentalis as a synonymous taxon due to lack differentiating diagnostic features. Sinclair (1921) differentiated H. mentalis from H. primaevus upon the former’s more pronounced genial flange and Chadron Formation occurrence. Scott & Jepsen (1936) followed this trend by limiting H. primaevus to the Brule Formation, and several subsequent authors followed suit (e.g., Simpson, 1941; Hough, 1949; Morea, 1975). Clark (1937) was the first to refer a specimen from the Chadron Formation to a synonym of H. primaevus (H. robustus), and Bryant (1996) reported on preliminary findings indicating additional Chadronian occurrences.

Figure 46 Bayesian cladistic analysis.

50% majority-rule consensus tree, generated from clade posterior values within the stationary pool. Values below nodes represent posterior probability support percentages.

This study’s cluster analysis (Fig. 43) placed specimens diagnosed to H. mentalis, utilizing historical criteria, throughout morphogroup three of the three-fold subdivision of the Hoplophoneus genus. This is indicative of a lack of clear morphometric boundaries between this species and that of H. primaevus. Furthermore, no differential character states were observed between specimens of H. mentalis and H. primaevus, which when combined leaves this study to conclude H. mentalis to be a junior synonym of H. primaevus. With the synonymy of these two taxa, the FAD of H. primaevus is extended to approximately 35.5 Ma, the lowest occurrence of “H. mentalis” material (Bryant, 1996). The subsequent DFA performed on the Hoplophoneus genus produced substantial support (95.12%) for a four-fold subdivision (as seen in Fig. 43). Two of these groups are allocated to the species of H. occidentalis and H. cerebralis. However, the remaining two emanate from a split within what has traditionally been perceived as a singular H. primaevus species. This study found numerous intraspecific variable character states for H. primaevus. Though none of these morphological features were found to align themselves within the DFA-supported morphogroups. So, likewise to the Dinictis genus, substantial qualitative intraspecific variation is present for this species, and the recovered metric differences seen in the DFA are likely sexual dimorphic, temporal or regional trends.

Hoplophoneus oharrai (Jepsen, 1926)

Referred taxa Hoplophoneus oharrai (Jepsen, 1926)

Type —SDSM 2417, skeleton

Referred specimens —none.

Distribution Late Chadronian (Ch3) of South Dakota (Chadron Fm., Crazy Johnson Mbr.).

Diagnosis Characters of Hoplophoneus plus: sutural contact absent between the lacrimal and jugal; medial ridge of palate absent; postglenoid foramen present; posterior lip of the glenoid socket projects more ventrally than the anterior lip; oblique angle between the braincase, disregarding the sagittal crest, and axial plane of the cranium; anterior dentary position elevated above cheek tooth row; vertically orientated posterior border of coronoid process; deep genial flange between 32 and 50% of total dentary length; mesial-distal length of C1 greater than that of P4; trigonid proportion of m1, 88% or greater.

Autapomorphic and descriptive features A nimravid of moderate size, with a basilar length of 187 mm, (n = 1); lambdoid crest angle of 122 degrees, (n = 1); C1 compression of 2.40, (n = 1); enlarged paroccipital process relative to other Hoplophoneus taxa; humerofemoral index higher (90%, (n = 1)) than that of H. primaevus, and H. occidentalis, which ranges from 80–87%, mean of 83.54%, (n = 13).

Discussion —Jepsen (1926) named Hoplophoneus oharrai after a mostly complete skeleton from the Crazy Johnson Member of the Chadron Formation, South Dakota (Clark, 1937). Diagnosing criteria for this species were cited as the deep genial flange, and relative and absolute proportions of the skeleton. Simpson (1941) tentatively synonymized this species with that of H. mentalis (Sinclair, 1921), a conclusion upheld by Hough (1949) eight years later. Both of these decisions were primarily made on the Chadronian co-occurrence of the species and shared, pronounced genial flange.

The type specimen of H. mentalis consists of a left dentary recovered from the Peanut Peak Member of the Chadron Formation, South Dakota (Clark, Beerbower & Kietzke, 1967). The findings of this study indicate that H. mentalis is a junior synonym of H. primaevus, exhibiting no differential features from that of H. primaevus. The type of H. oharrai however, exhibits several differential character states and metric peculiarities (such as the relative proportions of the appendicular skeleton and shape of the lower jaw) distinct from any other Hoplophoneus species, and thus is returned to valid status. However, being the only specimen this study could allocate to this species (limiting the use of the other metric analyses utilized in this study), the possibility remains of a spectrum existing which incorporates the current “natural gap” between H. oharrai and its most similar taxon, H. primaevus.

Radiometric ages have yet to be presented for the Crazy Johnson Member of the Chadron Formation in South Dakota. However, the end of the Chadronian is presented as 33.89 Ma (Vandenberghe et al., 2012). Therefore, for this study a conservative date of 34 Ma was selected as the FAD of this taxon.

Hoplophoneus occidentalis (Leidy, 1866)

Referred taxa

“Drepanodon”/“Machairodus” occidentalis (Leidy, 1866)	
Hoplophoneus occidentalis (Cope, 1879b)	
Dinotomius atrox (Williston, 1895)	

Type ANSP 11074, left dentary fragment with p4.

Referred specimens —UNSM 322-51, FAM 62075, FAM 102387, DICK 33-1370, FAM 62025, AMNH 1407.

Distribution —Orellan of Nebraska (Brule Fm., Orella Mbr.), North Dakota (Brule Fm.), South Dakota (Brule Fm., Scenic Mbr.), Wyoming (White River Fm., Brule Mbr.); Whitneyan of South Dakota (Brule Fm., Poleslide Mbr.).

Diagnosis —Characters of Hoplophoneus plus: sutural contact absent between the lacrimal and jugal; medial ridge of palate present; postglenoid foramen absent; posterior lip of the glenoid socket projects more ventrally than the anterior lip; oblique angle between the braincase, disregarding the sagittal crest, and axial plane of the cranium; anterior dentary position elevate above cheek tooth row; vertically orientated posterior border of coronoid process; deep genial flange between 32 and 50% of total dentary length; mesial-distal length of C1 less than that of P4; ratio of height between P3 and P4 0.50–0.70; p3 lower than p4; trigonid proportion of m1, 88% or greater.

Autapomorphic and descriptive features —A nimravid of large size with basilar length between 223–242 mm, mean of 231, (n = 3); lambdoid crest angle ranging from 109 to 118 degrees, mean of 113, (n = 3); serration density per millimeter of 4.1, (n = 1); C1 compression of 2.05, (n = 1); lacrimal process present.

Discussion —The first mention of Hoplophoneus occidentalis was in the minutes of an Academy of Natural Sciences of Philadelphia meeting, where Leidy (1866) characterizes “…a new sabre-toothed tiger, under the name of Drepanodon or Machairodus occidentalis, a species larger than its contemporary the D. primaevus” (p. 345). However, further description of this specimen did not occur until three years later (Leidy, 1869). Both of these publications cited size as the distinguishing features of this species from that of Drepanodon (=Hoplophoneus) primaevus, a view which was maintained by most subsequent authors (e.g., Adams, 1896b; Scott & Jepsen, 1936; Simpson, 1941; Morea, 1975). Williston (1895) named Dinotomius atrox after a nearly complete skeleton, a view which was challenged by Hough (1949), citing a composite of at least three individuals. Adams (1896a) synonymized Williston’s specimen(s) to H. occidentalis. Hough (1949) suggested that H. occidentalis may be a large subspecies of H. primaevus, though this view has not been entertained subsequently. Morea (1975) noted the similarities in proportions between H. occidentalis and the type of H. dakotensis, and suggested that the additional differential feature of absent p3 in the latter may be unreliable in distinguishing these species.

Simpson (1941) viewed H. occidentalis as a rare migrant to the H. primaevus dominant paleolandscape of the Brule Formation. Specimens of H. occidentalis are indeed rare and have been historically identified on their great non-overlapping metrical range with H. primaevus (lacking additional differential features). This study found statistical support for the validity of this species in its cluster and subsequent discriminant function analyses (Figs. 43 and 44 respectively). Additionally, this study identified an additional character in distinguishing this species, the absence of the postglenoid foramen. However, due to the shortage of specimens belonging to this taxon, especially those representing different ontogenetic stages, a complete picture of the variability of this character is yet unknown, nor is size range. Currently, it is less assumptive to categorize these specimens as a valid species than potential male members of H. primaevus, which would imply a female to male ratio around 25:1 based upon currently known fossil specimens from the Brule Formation.

Hoplophoneus sicarius (Sinclair & Jepsen, 1927)

Referred taxa

“Eusmilus” sicarius (Sinclair & Jepsen, 1927)	
Hoplophoneus sicarius (Morea, 1975)	
“Eusmilus” sicarius (sensuPeigné, 2003)	

Type —PU 12953, cranium, right dentary.

Referred specimens —None.

Distribution —Orellan of South Dakota (Brule Fm., Scenic Mbr.).

Diagnosis —Characters of Hoplophoneus plus: medial ridge of palate absent; postglenoid foramen absent; anterior and posterior lip of the glenoid socket project equally ventrally; oblique angle between the braincase, disregarding the sagittal crest, and axial plane of the cranium; anterior dentary position lies in same plane as cheek tooth row due to elevation of cheek teeth; forwardly orientated posterior border of coronoid process; extremely deep genial flange, 54% or more of total dentary length; mesial-distal length of C1 greater than that of P4; ratio of height between P3 and P4 0.45 and lower; p3 absent; trigonid proportion of m1, 88% or greater.

Autapomorphic and descriptive features —A nimravid of moderate size with basilar length of 189 mm, (n = 1); lambdoid crest angle of 96 degrees, (n = 1); lacrimal process present; extremely tall and laterally compressed C1s, length to width approximately 6.6:1; dorsal rotation of face and palate, relative to basicranium, creates an angle of approximately 20 degrees, this angle is closer to zero or slightly negative in most nimravid taxa.

Discussion —Sinclair & Jepsen (1927) assigned this species to Eusmilus due to a perceived similar dental formula and mandibular morphology of that of the Old world generic namesake E. bidentatus. Morea (1975) placed this species within Hoplophoneus, noting the differential cranial morphology between Old and New World species sharing this generic assignment. Specifically, Morea viewed the angle between the face and braincase as the most telling feature. Peigné (2003) returned H. sicarius back to the original Eusmilus designation, due to the returned topology in his phylogenetic analysis, specifically the proximity of it, E. villebramarensis and “E.” cerebralis.

The results of this study’s phylogenetic analyses render the Hoplophoneus genus paraphyletic, a conclusion explicitly discussed in Bryant (1996), and implicit in the tree of Peigné (2003). As previously discussed, this study advocates the disuse of Eusmilus, and all previously contained taxa (at least for North American nimravids) moved to the Hoplophoneus designation to maintain a monophyletic genus. This of course includes H. sicarius, and by doing so will hopefully end the volatile and unfruitful generic reassignment of this taxon.

The exact stratigraphic range of H. sicarius is poorly constrained. The type specimen originated from the Scenic Member, Brule Formation, South Dakota, with associative age of early Orellan (Bryant, 1996). However, barring additionally diagnosed specimens and more precise stratigraphic information, the described range of the entire Orellan is given.

Hoplophoneus dakotensis (Hatcher, 1895)

Referred taxa

“Eusmilus” dakotensis (Hatcher, 1895)	
Hoplophoneus dakotensis (Kretzoi, 1929)	

Type —YPM PU 11079, right dentary.

Referred specimens —UNSM 1068, SDSM 2815, SDSM 2830.

Distribution —Whitneyan of South Dakota (Brule Fm., Poleslide Mbr.).

Diagnosis —Characters of Hoplophoneus plus: sutural contact absent between the lacrimal and jugal; anterior and posterior lip of the glenoid socket project equally ventrally; oblique angle between the braincase, disregarding the sagittal crest, and axial plane of the cranium; anterior dentary position lies in same plane as cheek tooth row due to elevation of cheek teeth; vertically orientated posterior border of coronoid process; deep genial flange, between 32 and 54% of total dentary length; mesial-distal length of C1 greater than that of P4; ratio of height between P3 and P4 is 0.50–0.70; p3 absent; trigonid proportion of m1, 77–87%.

Autapomorphic and descriptive features —A nimravid of large size with basilar length of 238 mm (SDSM 2830) and 251 mm (SDSM 2815), Morea (1975); C1 compression of 2.25, (n = 1); extremely large and robust mastoid process; zygomata short and angular.

Discussion —There are no associated crania and dentaries for this taxon. Hatcher (1895) established the type on a right dentary lacking the p3. Jepsen (1933) assumedly referred the listed SDSM specimens based upon shared stratigraphic horizon with the type, large size, and similar extreme sabretooth developments as seen in H. sicarius and H. cerebralis. Kretzoi (1929) was the first (though tentatively) to refer this species to Hoplophoneus, but Jepsen (1933) cited shared morphological features with it and Eusmilus, specifically dental formula and several relative cranial proportions. Morea (1975) argued for a Hoplophoneus assignment for this taxon, noting differences between it and the “true” members of the Eusmilus genus (e.g., “E.” bidentatus, “E.” cerebralis). Even with this taxonomic assignment, Morea was tentative about the validity of this species, noting overall morphologic similarity between the type of H. dakotensis and specimens of H. occidentalis, differing only in the presence or absence of the p3. Morea held that the p3 was likely variable in H. dakotensis, and H. sicarius, but without additional specimens maintained this species delineating feature.

The whereabouts of SDSM 2815 (allotype of Jepsen, 1933) and 2830 are currently unknown, while UNSM 1068 was previously mentioned in literature as UNSM 6-12-7-95. The latter specimen and the type were thus the only specimens directly examined during the course of this study (F:AM 102387, referred to as H. dakotensis by Bryant, (1996) has been allocated to H. occidentalis). Even with Morea’s hesitancy regarding this species, the findings of this study determine it to be valid, primarily from information obtained from the referred crania, differing in several characters from that of the most closely allied species H. occidentalis and H. sicarius.

The type specimen and SDSM craniums originated from the Protoceras beds of the Poleslide Member of the Brule Formation in South Dakota. Jepsen (1933) inferred a similar origin for the UNSM specimen based upon preservation and adhered matrix. From this information the stratigraphic range of this species was inferred to occupy the latest Whitneyan, approximately 30.5–29.75 Ma (Bryant, 1996; Vandenberghe et al., 2012).

Hoplophoneus cerebralis (Cope, 1880a)

Referred taxa

“Machaerodus” cerebralis (Cope, 1880a)	
Hoplophoneus cerebralis (Cope, 1880b)	
Hoplophoneus belli (Stock, 1933)	
“Eusmilus” cerebralis (Toohey, 1959)	
Ekgmoiteptecela olsontau (Macdonald, 1963)	

Type —AMNH 6941, cranium.

Referred specimens —FAM 98189; FAM 69377; AMNH 98770; FAM 98769; YPM PU 16271; SDSM 54247; LACM 463.

Distribution —Late Chadronian (Ch3) of South Dakota (Chadron Fm., Crazy Johnson Mbr.); Whitneyan of South Dakota (Brule Fm., Poleslide Mbr.); early Arikareean (Ar1-2) of California (Sespe Fm.), Oregon (John Day Fm., Turtle Cove Mbr.), South Dakota (Sharps Fm.), and Wyoming (White River Fm.).

Diagnosis —Characters of Hoplophoneus plus: sutural contact present between the lacrimal and jugal; medial ridge of palate present; postglenoid foramen absent; anterior and posterior lip of the glenoid socket project equally ventrally; braincase, disregarding the sagittal crest, and axial plane of the cranium lie parallel, or nearly so, to each other; anterior dentary position lies in same plane as cheek tooth row due to elevation of cheek teeth; extremely deep genial flange, 54% or more of total dentary length; mesial-distal length of C1 is less than that of P4; ratio of height between P3 and P4, 0.45 and lower; p3 absent; trigonid proportion of m1, 88% or greater.

Autapomorphic and descriptive features —A nimravid of diminutive size with basilar length ranging from 94 to 100 mm, mean of 97 mm, (n = 4); lambdoid crest angle ranging from 92 to 113 degrees, mean of 99 degrees, (n = 5); serration density per millimeter of 6.2, (n = 1); C1 compression ranging from 1.73 to 2.48, mean of 2.13, (n = 4); lacrimal process present; anterior cusp of P3 variable; protocone of P4 commonly absent from fusion of anterior roots; extremely tall genial flange, approximately 73% of total dentary length (determined from FAM 69377, though possessing an incomplete dentary, the mostly complete cranium allows a likely accurate estimate).

Discussion —Cope (1880a) named Machaerodus cerebralis from a mostly complete cranium originating from the John Day Formation of Oregon, which he rectified to Hoplophoneus later that year (Cope, 1880b). Stock (1933) established Hoplophoneus belli on a severely crushed juvenile cranium from the Sespe Formation of California, to which distinguishing features were given as smaller size, more slender canines, less-developed parastyle of P4, and greater overlap of the P3 and P4 than compared to H. cerebralis. Hough (1949) held H. belli as a potential subspecies of H. primaevus, residing in California, but was tentative until additional specimens could be found. Toohey (1959) concluded that H. cerebralis and H. belli were congeneric, but posited their allocation to the Eusmilus genus based upon comparisons to E. bidentatus. Macdonald (1963) named Ekgmoiteptecela olsontau from a right partial dentary containing p4-m1. At the time no lower jaw material was known for specimens referable to H. cerebralis or H. belli. Macdonald held that his species and two aforementioned taxa were congeneric, but that they should be differentiated from Hoplophoneus and Eusmilus and referred to Ekgmoiteptecela. Morea (1975) synonymized H. belli with “Eusmilus” cerebralis, believing the type specimen of the former species derived its distinctiveness from its immaturity and individual variation. Additionally, Morea held Ekgmoiteptecela and Eusmilus to be congeneric, but stopped short of synonymizing E. olsontau and E. cerebralis, for associated craniums and jaws were still unknown.

Specimen LACM 5465 (referred to E. olsontau by Morea, 1975) was unable to be examined during the course of this study, but metrical values taken from that publication seem to indicate that the specimen may belong to a species distinct from the diagnosed H. cerebralis of this study. The aforementioned specimen is described with a genial flange height to mandibular length ratio of approximately 45% (compare to FAM 69377 and 73%). Future examination of this specimen will be required to determine its specific allocation.

Specimen YPM PU 16271 consists of an edentulous left dentary which compares well to the additional referred specimens. The stratigraphic occurrence of this specimen is the base of the Crazy Johnson Member, Chadron Formation, South Dakota. If correctly referred, this would extend the FAD of this taxon back by at least 4 Ma. As already discussed, the radiometric ages are poorly known for Crazy Johnson Member. Therefore, like the FAD of H. oharrai, a conservative date of 34 Ma was selected as the FAD of this taxon.

Genus Nanosmilus (Martin, 1992)

Type and only referred species —Nanosmilus kurteni (Martin, 1992)

Distribution —Orellan of Nebraska (Brule Fm., Orella Mbr.).

Diagnosis —Sutural contact between the lacrimal and jugal absent; petrobasilar and posterior lacerate foramina form two distinct grooves; large tabular mastoid with reduced to near absent paroccipital process; posterior lip of the glenoid socket projects more ventrally than anterior lip; angle between the braincase, disregarding the sagittal crest, and the axial plane of the cranium is oblique; postorbital process of frontal projects ventrally; anterior dentary position is elevated above cheek tooth row; vertically orientated posterior border of coronoid process; absence of fossa on ventral face of chin; incisors all caniniform, i1 very transversely compressed and i3 nearly as large as the lower canine; ratio of crown height between P3 and P4 is 0.50–0.70; parastyle on P4; P4 protocone reduced, short, crest-like or absent; M1 transversely reduced, crest-like, with low cusps and near absent to absent protocone; lower incisor arcade curved; p3 lower than p4; anterior cusp on p4 mesially-distally shorter than the posterior cusp; the main cusp of the p4 is as tall or taller than the paraconid of m1; m1 metaconid absent; trigonid proportion of m1 88% and higher; serrations present on adult minimally worn cheek teeth.

Discussion —Martin (1992) held Nanosmilus as an early form in the Eusmilus lineage (Eusmilini), not including “E.” sicarius and “E.” dakotensis. Martin argued for several synapomorphies which differentiated his Eusmilini from the Hoplophoneus genus: possession of a parastyle on the P4, reduction of talonid on the m1 to a basal projection, low sagittal crest, shortened nasals, extreme anterior placement of the anterior palantine canal, and small size. Bryant (1996) showed that all of these features either occurred consistently or variably in Hoplophoneus, or were products of the juvenile nature of the type specimen, thus Nanosmilus was tentatively held as synonymous to Hoplophoneus.

The returned phylogenies of this study place the type specimen of Nanosmilus at the base of a well-supported clade containing the entirety of the Hoplophoneus genus. However, Nanosmilus presents a set of differential character states which set it apart from all species of the aforementioned genus: the main cusp of the p4 is as tall as or taller than the paraconid of m1; metaconid of m1 absent. Therefore, pending future work and acquisition of additional specimens, the genus of Nanosmilus is held as valid.

Nanosmilus kurteni (Martin, 1992)

Type —UNSM 25505, cranium and dentary of a sub-adult individual (C1 not erupted).

Referred specimens —None.

Distribution —Same as genus.

Diagnosis —Same as genus.

Autapomorphic and descriptive features —A nimravid of diminutive size with basilar length of approximately 107 mm, (n = 1); lambdoid crest angle of 120 degrees, (n = 1); lacrimal process absent; anterior cusp on P3 and p3.

Discussion —Martin (1992) established Nanosmilus kurteni on a sub-adult (unerupted permanent C1s) cranium and dentary from the Orella Member of the Brule Formation, Nebraska. Among the distinguishing features noted were the small size, narrow skull, sagittal crest forming a “V” above the glenoid fossa, premaxillaries ending well-posterior to the posterior edge of C1, small genial flange, and optic foramen and orbital fissure separate. Bryant (1996) regarded N. kurteni as a valid species in his cursory revision of the North American Nimravidae, but suggested future synonymy with the Hoplophoneus genus with direct study of the specimen.

UNSM 25505 was returned as a valid species for this study with character states differentiating it from both the Hoplophoneus genus and potential Dinictis-like ancestor alluded to by Martin (1992). The extent to which ontogeny factors into the preserved features of this specimen may change the future generic allocation of this taxon with the discovery of additional specimens, however, the absence of the metaconid on the m1, and size of the p4 relative to the m1 at least indicate a species distinct from that of currently defined Hoplophoneus, while the suite of advanced saber-tooth morphology indicate a taxon more derived than Dinictis, or Pogonodon.

The exact stratigraphic occurrence of the type specimen is currently unknown beyond the Orella Member of the Brule, Nebraska, thus pending better constraint, the stratigraphic range was decided to be the entire Orellan for this study.

NIMRAVINI Cope (1880b)

Definition —The most recent common ancestor of Nimravus brachyops, N. intermedius, Dinaelurus crassus, and all of its descendants.

Included Genera —Nimravus, Dinaelurus.

Diagnosis —Synapomorphies: deep masseteric fossae on the medial and lateral faces of the zygomata; fossa on the ventral surface of the chin; spatulated incisors, with accessory denticules especially on the lower incisors, I3 slightly caniniform and distinctly larger than the other incisors; anterior cusp on p4 mesially/distally longer than the posterior cusp; serrations absent on adult minimally-worn cheek teeth.

Discussion —This clade was returned in all cladistic analyses, with substantial support in the Bayesian and parsimony analyses. Like the Hoplophoneini clade, both Bryant (1996) and Peigné (2003) returned similar groupings. The Nimravini clade presents as plesiomorphic within the Nimravidae, with features more generalized and analogous to modern felids than that of Dinictis, Pogonodon, and the Hoplophoneini.

Genus Nimravus (Cope, 1879b)

Referred taxa

“Machaerodus” (sensuCope, 1878)	
“Hoplophoneus” (sensu (Cope, 1879a)	
Nimravus (Cope, 1879b)	
Archaelurus (Cope, 1879c)	
“Pogonodon” (sensuCope, 1880b)	
“Dinictis” (sensuAdams 1896b )	

Type species —Machaerodus brachyops (Cope, 1878)

Additional referred North American species —None.

North American distribution —Whitneyan of Nebraska (Brule Fm., Whitney Mbr.), South Dakota (Brule Fm., Poleslide Mbr.); late Whitneyan or early Arikareean of Saskatchewan (Cypress Hills Fm.); early Arikareean (Ar1) of California (Sespe Fm.), Nebraska (Gering Fm.), Oregon (John Day Fm., Turtle Cove and Kimberly Mbrs.), Wyoming (Arikaree Grp.).

Diagnosis —Sutural contact between the lacrimal and jugal; presence of lateral and medial fossae on the zygomata; broadly circular zygomata in dorsal view; presence of discrete petrobasilar and posterior lacerate foramina; postglenoid foramen present; reduced mastoid with large plate-like paroccipital process; posterior lip of the glenoid socket projects more ventrally than anterior lip; oblique angle between the braincase and axial plane of the cranium; horizontally projecting postorbital process of frontal; posteriorly orientated posterior border of coronoid process; no genial flange, but the ventral rim of the chin is distinctly angulate; presence of fossa on ventral face of chin; spatulated incisors, with accessory denticles especially on the lower incisors, I3 slightly caniniform and distinctly larger than the other incisors; mesial-distal length of C1 less than that of P4; ratio of height of P3–P4, 0.71 and greater; parastyle absent from P4; P4 protocone reduced, short, crest-like; M1 transversely reduced, crest-like, with low cusps and near absent to absent protocone; lower incisor arcade not or little curved, so as i1 is not visible in lateral view; p3 height is as tall or slightly taller than p4; anterior cusp on p4 mesially/distally longer than the posterior cusp; p4 as tall as or taller than the paraconid of m1; m1 metaconid absent; trigonid proportion of m1 is 77–87%; serrations absent on adult minimally worn cheek teeth; ratio of tibia to femur 87% and higher; articulation between the calcaneum and navicular present.

Discussion —Cope (1880b) held that the genera of Archaelurus and Nimravus, referred to as the “false sabre-tooths,” formed an intermediate connection between the primitive cats (e.g., Pseudaelurus) and the primitive sabre-tooths (e.g., Dinictis, Pogonodon, and Hoplophoneus). Scott & Jepsen (1936) maintained this view by placing Nimravus within its own Subfamily of the Felidae, and representative of a transitional form between “true cats” and machairodontines (e.g., Eusmilus, Smilodon). The intermediate status of Nimravus has subsequently been rejected with in-depth reviews of the anatomy of nimravids, felids, and associated groups (Baskin, 1981; Bryant, 1988; Hunt, 1987; Neff, 1983). However, within the currently defined Nimravidae, a basal position has been consistently returned for Nimravus in cladistic analyses (Bryant, 1996; Peigné, 2003). This study is no different in returned relationships, and therefore would seem to indicate that Nimravus may approximate the morphology of an ancestral nimravid.

Nimravus brachyops (Cope, 1878)

Referred taxa

“Machaerodus” brachyops (Cope, 1878)	
“Hoplophoneus” brachyops (Cope, 1879a)	
Nimravus brachyops (Cope, 1879b)	
Archaelurus debilis (Cope, 1879c)	
Nimravus gomphodus (Cope, 1880b)	
Nimravus confertus (Cope, 1880b)	
“Pogonodon” brachyops (Cope, 1880b)	
“Dinictis” brachyops (Adams, 1896b)	
Dinictis major (Lucas, 1898)	
Archaelurus debillis major (Merriam, 1906)	
Archaelurus debillis merriami (Hay, 1929)	
Nimravus sectator (Matthew, 1907)	
Nimravus meridianus (Stock, 1933)	
Nimravus bumpensis (Scott & Jepsen, 1936)	
Nimravus altidens (MacDonald, 1950)	

Lectotype —AMNH 6935, partial right dentary.

Referred specimens —AMNH 6930, USNM 3957, SDSM 55280, SDSM 521, SDSM 5776, SDSM 15012, SDSM 348, FAM 62151, AMNH 6931, YPM 10045, YPM 1044, YPM 10517, YPM 10046, YPM 14388, YPM 10519, YPM 14385, AMNH 6936, AMNH 6933, AMNH 12882.

North American distribution —Same as genus.

Diagnosis —Same as genus.

Autapomorphic and descriptive features —A nimravid of moderately large size with basilar length ranging from 159 to 194 mm, mean of 177 mm, (n = 9); lambdoid crest angle ranging from 133 to 146 degrees, mean of 138 degrees, (n = 10); serration density per millimeter ranging from 2.2 to 2.3, mean of 2.2, (n = 4); C1 compression ranging from 1.56 to 1.84, mean of 1.74, (n = 11); lacrimal process absent; alveolar torus variable; anterior P3 cusp absent; anterior p3 cusp variable.

Discussion —Cope (1878) established Machaerodus brachyops on a hypodigm of specimens including: one complete cranium, two partial crania, a “left mandibular ramus” and associated skeleton, and several isolated teeth. Cope diagnosed his species upon: the ratio of interspace length between the P2-P3/C1-P2, large size of the premolars, absent anterior cusp of p3, and an angulate ventral border of the dentary. By the time Toohey (1959) performed his revision of the genus, an additional six species and two subspecies had been added. Toohey established the lectotype of AMNH 6935, a partial right dentary, for N. brachyops, viewing it as the only specimen from the Cope collection to agree with the description in the hypodigm, and that the cited “left mandibular ramus” was a lapsus calami. Additionally, Toohey concluded that all of the above named taxa formed a spectrum of spatiotemporal variation and sexual dimorphism of a single species, though additional hypotheses were presented allotting for differential subspecies or species through time and region.

Toohey (1959) could not distinguish any consistent morphological differences from Nimravus brachyops and Nimravus intermedius (of Europe), leaving only temporal and geographic criteria as their distinguishing features. Peigné (2003) demonstrated that N. intermedius displayed a greater range in measurements of the m1 than that of N. brachyops, but that they almost entirely overlapped on the higher end of those values. Peigné held differences between the two species in N. brachyops’ larger size, loss of the p1, and rarely present m2. This study found no differential character states between the two taxa (dissregarding missing data), and viewed the p1 and m2 as traits, not characters, but refrains from concluding them synonymous pending additional morphometric analysis of european specimens.

The alveolar torus has long been noted as a diagnostic feature of this taxon, first described by Cope (1879d) as a provisional growth to strengthen the dentary at the area of greatest strain. Toohey (1959) correlated the feature to ontogenetic and geologic age, with less mature and geologically older individuals possessing smaller examples of the growth, but could not conclude a definitive function. Peigné (2003) demonstrated that no clear association between tooth wear (a proxy for ontogenetic age) and torus growth existed in the available specimens, but did exhibit a propensity for John Day specimens to display the feature over Great Plains examples regardless of tooth wear stage. The purpose and geographic distribution of the alveolar torus is beyond the scope of this study, though presence and absence of the feature was found in both morphogroups recovered from the performed cluster analysis of this study (Fig. 40). Nevertheless, the structure still functions as a potentially useful diagnostic feature for this taxon, being absent in all other species of nimravid.

This study arrived at the same conclusions of synonymy as did Toohey (1959). Two main morphogroups were recovered from the performed cluster analysis (Fig. 40) though the subsequent discriminant function analysis returned very poor support for these groups (33.33% ), implying only one contained species for the North American specimens examined. However, two specimens, AMNH 6933 and SDSM 55280, pose potential problems on the differential character state diagnoses thus far presented. Both of these specimens possess either inaccessible or incomplete basilar cranial measurements, though being assured of very large size, and potentially existing in the second state of character one (BL length of 205 mm and over). SDSM 55280 lacks a preserved basicranium, but Cope (1883) gave a size estimate of 206 mm for AMNH 6933, and the SDSM specimen is comparable in size. Whether these specimens represent exemplar male specimens of this species (as concluded for AMNH 6933 by Toohey (1959)), or distinct taxa will require additional specimens or more in depth morphological analyses.

Genus Dinaelurus (Eaton, 1922)

Type and only referred species —Dinaelurus crassus (Eaton, 1922)

Distribution —Early Arikareean (Ar2) of Oregon (John Day Fm., Turtle Cove Mbr.).

Diagnosis —Presence of lateral and medial fossae on the zygomata; zygomata triangular in dorsal view; absence of medial ridge of palate; presence of discrete petrobasilar and posterior lacerate foramina; postglenoid foramen present; reduced mastoid with large plate-like paroccipital process; anterior lip of the glenoid socket absent; oblique angle between the braincase and axial plane of the cranium; horizontally projecting postorbital process of frontal; spatulated incisors, with accessory denticules, I3 slightly caniniform and distinctly larger than the other incisors; mesial-distal length of C1 less than that of P4; no serrations on adult C1; ratio of height of P3–P4, 0.71 and greater; parastyle absent from P4; P4 protocone reduced, short, crest-like; M1 transversely reduced, crest-like, with low cusps and near absent to absent protocone; serrations absent on adult minimally worn cheek teeth.

Discussion —Eaton (1922) viewed Dinaelurus as a taxon more advanced than Nimravus with developments toward the Felidae, though no phylogeny was given. Specifically noted were the development of the otic bullae, shortened length of the alisphenoid canal, and the slender proportions of the alisphenoid bone. Similarly to Eaton, De Beaumont (1964) held Dinaelurus’ phylogenetic affinity closer to Nimravus, as did Martin (1980) who erected a monogeneric tribe (Dinaelurini) to contain it. Martin viewed Dinaelurus as a “conical-toothed cat” derived from a Nimravus-like ancestor in his suggested phylogeny. Bryant (1996) and Peigné (2003) were the first to perform cladistic analyses on the Nimravidae, and returned hypotheses evocative of previous authors, with a returned Nimravini clade (Nimravus, Dinaelurus), however Peigné (2003) found poor internal resolution for this clade, and attributed it to fragmentary knowledge of Dinaelurus.

This study also returned a well-supported Nimravini clade in all of its cladistic analyses, with a basal arrangement of Dinaelurus relative to the species of N. brachyops and N. intermedius. However, due to the imperfectly known character states of Dinaelurus, particularly those belonging to the dentary and postcrania, differential relationships may be returned in future analyses with the presence of new material. Though pending, the discovery of such material the generic validity of this taxon is maintained, largely due to its uniqueness, being the only non-sabretooth nimravid, and associated preserved character states.

Dinaelurus crassus— (Eaton, 1922)

Type —YPM 10518, cranium.

Distribution —Same as genus.

Diagnosis —Same as genus.

Autapomorphic and descriptive features —A nimravid of moderate size with basilar length of 174 mm, (n = 1); lambdoid crest angle of 126 degrees, (n = 1); ridge on posterior edge of C1, but lacking in serrations; cranium extremely broad for its length (164:174 mm (Eaton, 1922)); lacrimal process absent; ellipsoid C1 cross-section with compression ratio of 1.4; anterior P3 cusp absent.

Discussion —(Eaton, 1922) named Dinaelurus crassus upon a mostly complete cranium from the John Day Formation, Oregon. Though presenting the basicranial synapomorphies of the Nimravidae, the type specimen remains the only nimravid not to present saber-tooth dentition and its associated cranial adaptations. Eaton expanded upon this by characterizing the dentition as being more truly feline than other known nimravid taxa. Martin (1980) viewed the domed cranium and enlarged internal nares as potential cheetah-like adaptations, and organized the species as representative of a “conical-toothed cat.” The C1 is slightly laterally compressed with a reported value of 1.4 (Eaton, 1922), while typical values for nimravids, such as H. primaevus, range from approximately 1.5 to 2.3. Bryant (1996) noted the enamel ridge on the posterior edge of C1 (a feature present in some felid taxa (King, 2012)), but refrained from concluding lack of serrations in life due to wear and taphonomic forces.

Eaton (1922) assigned the stratigraphic origin of the type specimen to the upper John Day Formation based upon the preservation, and associated matrix of the specimen. Bryant (1996) concluded an occurrence of early Arikareean, but even with the subsequent work of Bryant & Fremd (1998), the exact stratigraphic age of the type specimen is not well constrained. For this study, an occurrence from the start of the Arikareean to the end of the Oligocene was selected pending additional examples of this taxon or more precise stratigraphic information for the type.

Species of Uncertain or Indeterminate Status

Hoplophoneus strigidens (Cope, 1878)

Referred taxa

Machaerodus strigidens (Cope, 1878)	
Hoplophoneus strigidens (Cope, 1880b)	

Type —AMNH 6942, medial fragment of an upper canine.

Referred specimens —None.

Distribution —Early Arikareean (Ar1 or 2) of Oregon (John Day Fm.).

Autapomorphic and descriptive features —significantly compressed canine with ratio of approximately 3.3 (compare to H. primaevus with ranges from 1.55 to 2.35); exhibits shallow central grooves on the medial and lateral faces of the canine which contribute to an elongate hexagonal cross-section; extremely finely serrated on the mesial and distal edges, serration density per millimeter of 8.7.

Discussion —Cope suggested that this tooth represented an animal of proportions similar to either H. primaevus or H. cerebralis (Cope, 1878; Cope, 1883). Though, Adams (1896b) noted that the specimen was characterized by no features which made it referable to the Hoplophoneus genus, nor any other. The shallow grooves may represent a deciduous canine, yet this is only ever seen on the medial side, not both. “Hoplophoneus”strigidens is probably not a nimravid, but instead possibly a barbourofelid based upon these depressions on both faces of the canine, a feature present in all barbourfelids (Bryant, 1988). The implications of the extremely high serration density of the type specimen, and this possible relationship will require further research, for typical values of serration densities of barbourofelids seem to have yet been reported, though Martin (1980) suggested they to be high. If true, this would at least provide tentative evidence for a temporal and range extension from the Clarendonian to the Arikareean, or approximately 11Ma (Tseng, Takeuchi & Wang, 2010). However, the exact stratigraphic level of origin of the type specimen within the John Day Formation is unknown, potentially shrinking this temporal reassignment (Bryant, 1996).

Conclusion

The results of this study suggest twelve valid species of North American nimravid. These taxa are determined to belong to six monophyletic genera: Dinictis, Pogonodon, Nimravus, Dinaelurus, Nanosmilus and Hoplophoneus. Linear morphometric analyses and qualitative character assessment found Dinictis to be monospecific (D. felina), while Pogonodon to contain two valid species (P. platycopis and P. davisi). Similarly to Toohey (1959),Nimravus was determined to be monospecific for North America. Cladistic analysis supports the validity of Nanosmilus by the taxon presenting a differential suite of character states distinct from Hoplophoneus or any other genus. Hoplophoneus mentalis was found to be a junior synonym for H. primaevus, for diagnosable criteria of the type providing no differentiation from that of H. primaevus. However, the validity of Hoplophoneus oharrai is reinstated. While previously synonymized with H. mentalis, the type of H. oharrai presents several character states distinct from either of these taxa.

Through cladistic analyses Eusmilus is determined to represent a non-valid genus for North American taxa, suggesting non-validity for Old World nimravid species as well. This determination derives from the resultant tree topology of all cladistic analyses presenting a single monophyletic clade for all Hoplophoneus and Eusmilus taxa. To continue utilizing the Eusmilus genus would result in a paraphyletic Hoplophoneus genus, and since Hoplophoneus has precedence, it is retained as the valid designation.

Finally, two main clades with substantial support were returned for all cladistic analyses, the Hoplophoneini and Nimravini. The former tribe containing the genera of Nanosmilus and Hoplophoneus, while the latter tribe that of Nimravus and Dinaelurus. Ambiguous positions relative to these main clades were recovered for the European taxa: Eofelis, Dinailurictis bonali, and Quercylurus major; and the North American taxa Dinictis and Pogonodon.

Supplemental Information

Supplemental Information 1 List of specimens examined

Click here for additional data file.

Supplemental Information 2 Depictions of how specimens were measured

Click here for additional data file.

Supplemental Information 3 Raw measurement data

Click here for additional data file.

Supplemental Information 4 DFA coefficients

Click here for additional data file.

Supplemental Information 5 R code utilized in analyses

Click here for additional data file.

Supplemental Information 6 Modified data for R analyses

Click here for additional data file.

Supplemental Information 7 Character matrix

Click here for additional data file.

Supplemental Information 8 MrBayes code

Click here for additional data file.

This work stems, in part, from the work conducted during my Master’s thesis at South Dakota School of Mines & Technology, and as such I would like to thank the people and institutions which helped facilitate its completion. Firstly, Master’s committee members Dr. Darrin Pagnac (major advisor), and Dr. Clint Boyd. Secondly, the following people provided access to, and assistance with, the specimens examined during the course of this study: Ms. Samantha Hustoft (Museum of Geology, South Dakota School of Mines & Technology), Mr. George Corner (University of Nebraska Lincoln State Museum), Ms. Judy Galkin (American Museum of Natural History), Mr. Daniel Brinkman (Yale Peabody Museum of Natural History), Ms. Amanda Millhouse (Smithsonian, National Museum of Natural History), and Ms. Megan Cherry (Badlands National Park). Finally, I would like thank the reviewers of this manuscript who greatly improved its quality.

Additional Information and Declarations

Competing Interests

Author Contributions

Data Availability

The author declares there are no competing interests.

Paul Z. Barrett conceived and designed the experiments, performed the experiments, analyzed the data, contributed reagents/materials/analysis tools, wrote the paper, prepared figures and/or tables, reviewed drafts of the paper.

The following information was supplied regarding data availability:

Data set is provided in Supplemental Information.

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
