# Peer review of "Taxonomic and systematic revisions to the North American Nimravidae (Mammalia, Carnivora)"

_PeerJ, doi:10.7717/peerj.1658_

## Round 0.1 · original submission · Major Revisions

You present interesting phylogenetic analyses of the Nimravidae, which seems to be a rather problematic group based on limited material. Based on the mixed reviews and my own observations, I feel major revisions are in order. You use new methods which has so far not applied and do the first comprehensive analysis of European and North American taxa, which has so I far as I understand not been done making this a novel study in line with PeerJ´s publishing requirements. However, I partially agree with reviewer two that a major rewrite is necessary which takes better into account the strengths and weaknesses of your approach(es). I think the study is valuable in demonstrating that there is still too limited data is available to adequately or further resolve the phylogeny of Nimravidae. The paper should therefore be rewritten to test if the current data is sufficient to resolve the phylogeny of Nimravidae rather than the state of the art method to resolve its phylogeny once and for all. Publication in PeerJ is not pending upon results of high impact, but based upon new studies using adequate methologies and honest interpretations. From this perspective your paper can be still accepted for publication if the following major points are adequately addressed in the revision:

Phylogenetic Methods: you identify species by grouping methods and subsequently test these assignments by discriminant analysis in metric features. Reviewer 2 points out that this usually done the other way round and that both methods might not be complementary with the used species concept. I agree that it is not entirely clear how these methods can be combined with a modern species concept. I particularly wonder how the limited sampling and differences in sampling of different taxa influence the results. Taxonomy of fossil taxa based on morphology will probably always be subjective to some degree. You need to justify why and how your methods would make the process more objective.

Fossil record: It might be worthwhile to stress and adequately explain with relevant citations, why the fossil record of Nimravidae is comparatively poor. This does not only justify your limited dataset, but might also partially explains the problems encountered by other studies to resolve the phylogeny of this group. With the data you present, you should not focus on resolving the phylogeny of Nimravidae, but rather on testing if the current available data is enough to resolve the phylogeny of this group.

Nomen dubium: This term seems to be used in the wrong context for Hoplophoneus mentalis (see comments by reviewer 2)
Paleontological sections: Please homogenize the diagnoses of the taxa (see comments by reviewer 2)

Figures: I think it would help if there is a figure illustrating the main characters in the main text (see also comments by reviewer 2) as well as having schematic drawings of the analyzed taxa in one of the phylogenetic trees to better be able to follow the discussion for outsiders

In addition to the additional suggestions made by the reviewers, please also address:

Line 258-259: Be more specific: Were you as author unable to examine Dinaelurus crassus or was it unavailable for other reasons.

Line 429: additional, novel studies on tip-dated phylogenies are available, e.g., Arcila et al. 2015

Arcila, D., Alexander Pyron, R., Tyler, J. C., Ortí, G., and Betancur-R, R., 2015, An evaluation of fossil tip-dating versus node-age calibrations in tetraodontiform fishes (Teleostei: Percomorphaceae): Molecular Phylogenetics and Evolution, v. 82, Part A, p. 131-145.

·

Basic reporting

This manuscript is well-written and extremely thorough. It is clear to me that the author did a fine job of scrutinizing and measuring each nimravid specimen that he studied. I believe this article conforms to all measures of PeerJ policy. The format deviates slightly from the basic science article format, but I believe that this type of phylogenetic revision requires that.

Experimental design

I do not study phylogenies and I am ignorant of the latest techniques, therefore I am not in a great position to comment on the appropriateness of the techniques used to create the new tree, however, I am comfortable with the care taken in examining every specimen. It also appears that the author has recent phylogenetic techniques and has taken into account the most recent previously published phylogenies of the Nimravidae.

Validity of the findings

These data all appear to be robust and well-collected.

Additional comments

This is a very good paper, I am not the best person to review the phylogenetic techniques but it is clear that extreme care was taken when examining this clade. There are a few minor typos I caught. Line 32: "valid" should come before "nimravid" here. Line 399: Delete "jr." after Hunt.

·

Basic reporting

Overall, this paper adheres to the standards of basic reporting. The introduction and background are adequate and the structure of the paper is a fairly standard one for papers dealing with phylogeny. As someone interested in the morphology of these animals, I might have chosen to include at least some of the character illustrations now in supplementary in the main document and put the DFA results in the supplementary instead. But that is a subjective choice. The paper seems self-contained, although there is very little actual character analysis included apart from the illustrations in supplementary. This results in two things: 1) it isn’t clear how the data set relates to those of Bryant and Peigné; 2) it isn’t entirely clear how new characters (if any) were identified.
The presentation is of variable quality. The illustrations are adequate, but not more. The text sometimes loses itself in linguistic convulsions and at times becomes rather ambiguous. It could be greatly simplified for increased readability. The introduction includes a number of things, such as the discussion of the four character criteria, that are typical thesis text and should be shortened or removed.
Some specific points on this section are as follows. 1) The description of the analyses of Bryant and Peigné makes it appear as if they used different methodologies for their cladistic analysis. However, the the ie (implicit enumeration) procedure of Hennig86, used by Peigné, is computationally identical to the branch-and-bound implementation in in PAUP, used by Bryant. 2) Lines 165ff: The reason why nimravids are rare in the fossil record has nothing to do with their social system (which is unknown). The reason why so few specimens are known from any one site is that they were top predators, and therefore rare in the ecosystem. 3) Lines 203 and 207: Ontogeny and ontology are two quite different things. 4) Lines 230ff: Autapomorphies are essential to diagnose terminal taxa (i.e. the species here entertained) and so should not be eliminated.

Experimental design

In terms of ‘experimental’ design (not really experiments, but the analogous step) the paper is overall adequate. The goals are clearly stated. I think, without having tried the experiment, that I could reproduce the analyses if necessary, so the methods are, in my opinion, adequately described. Whether they are appropriate is another question (see below).

Validity of the findings

The major problem with this paper lies in the methodology, which the author makes much of. It seems to me that the author is trying to have it both ways: applying a ‘modern’ species concept (PSC) yet in the end identifying species by tried and true grouping methods. Thus, the author identifies ‘characters’ potentially exclusive states (the lacrimal process is mentioned). Then he does a discriminant analysis (DFA) to identify if the specimens with each state also differ in metric features. If no they are considered a single taxon if yes they are two or more taxa (depending on the number of states). Traditionally it would be the other way around: first some analysis grouping taxa by metrics is performed (regression or principal components are commonky used), then an attempt is made to identify characters within the groups identified by the metrics. Both approaches have problems since in reality there is no objective way to identify fossil species by morphology. I see some problems with the author’s approach. First, if the PSC were applied strictly the DFA shouldn’t be used – the different character states create diagnosable units and should thus be considered separate taxa. The author realises that this is not a realistic approach, hence the DFA. However, the latter is based on woefully small data sets, as well as on rules of thumb that may or may not be relevant to distinguishing between taxa among Nimravidae. In particular, the measurement set is very small for this task, and combined with the small number of specimens in the a priori groups I am unsure that the results are very meaningful. A further complication is that I cannot see that the equations for the LD1s in Figs 4, 6, 7, and 9 are provided. It is possible to run the analyses again, but this should not be required for understanding the graphs. If these functions are primarily size-driven, then they are largely irrelevant to taxon identification, unless the author wants to commit the same error as some of the early authors he criticizes, i.e., to identify species on the basis of small differences in size. (Of course, it is not the author’s fault that the data are limited, but this is what creates the difficulties with generating a phylogeny of the group in the first place.)
The systematic paleontology section has numerous problems and inconsistencies. The subsections vary from taxa to taxa; in some taxa the differential diagnosis says ‘same as genus’ when there is no differential diagnosis for the genus (‘diagnosis’ and differential diagnosis’ are two different things); some taxa have differential diagnoses, some do not; some taxa have ‘differential diagnoses’ and ‘autapomorphic and descriptive diagnoses’, others just ‘diagnoses’.
The term nomen dubium is incorrectly apploed to Hoplophoneus mentalis. Nomen dubium refers to a taxon, the material of which lacks entirely the diagnostic features of the group to which is assigned, or whether it is unknown if it has them (e.g., in the case of a lost holotype). In the present case, H. mentalis has the diagnostic that place it in Hoplophoneus, but no characters to differentiate it from H. primaevus, thus making the latter a subjective junior synonym of the latter.

In summary, although there is interesting information in this paper, it does not represent a substantial step forward compared to the analyses of Bryant and Peigné, in that the same groups are identified with confidence (Nimravini and Hoplophoneini and some genera) while the same problems persist. I do not feel that it can be published in its present form, but I also don’t feel that it is without merit. I would recommend a complete rewrite with reconsideration of the strengths and weaknesses of the approach, added consistency, and resubmission as a new manuscript.

---

## Round 0.2 · Minor Revisions

Thank you for implementing our suggestions. The manuscript has significantly improved and as good as accepted pending some minor additional points which need to be addressed. Note that these and other changes cannot be made anymore once the manuscript is accepted for publication.

The most important points to be addressed are:

Number of variables in a DFA: it should be pointed out somewhere that the number of variables in a DFA cannot exceed the number of cases within the smallest sample (see comments by the reviewer)
Use of tip-dating method on pure morphological data: I known tip-dating methods are commonly run without sequence data before the actual analysis, but unsure if it is meaningful to do it if no sequence data is available? Please confirm this by citing some references which have done so or at least say that is a meaningful analysis to do (compare also line 361-363)
Line 102: please replace “stem” with “might stem” or “probably stem” as this is an interpretation
Line 189: “not an” is exaggerated as all this remains to some degree subjective; please replace with “ less on”
Line 301, 302, 307: nothing is listed below “Cranial”, “Mandibular” and “Postcranial”; please briefly describe these characters if necessary or rearrange this
Line 361-363: I am aware of these methods, but I didn´t know they could be meaningfully applied without molecular sequence data. These methods are commonly run without sequence data before the actual analysis, but unsure if it is meaningful to do it if no sequence data is available? Could you maybe confirm this by citing some references which have done so or at least say so.
Line 372: replace “occuerence” with “occurrence”
Line 548 - : shouldn´t “m2” and “p3” be replaced with “M2” and “P3”? Throughout the systematic part teeth formulae are not consistently written in capitals, please verify this throughout the manuscript as I have noticed this in many more cases (line 595, etc.).
Line 575-578: This “Referred Taxa” is a bit confusing ? I guess you mean that species belonging to these genera are occasionally included in Pogonodon. If these are synonymous, why is the taxon not called Hoplophoneus as this would be the senior synonym? It becomes further unclear as Hoplophoneus is referred to subsequently on its own. If you mean that part of the species belong to it, wouldn´t it make more sense to use “partim”? Even be better would to leave “referred taxa” away below genera unless the type species of this genus belongs to the genus the taxon is referred to.
Table 3: For completeness sake, you would have the list the original reference listed in the paleobiology database with the FAD and LAD.

·

Basic reporting

The text and illustrations are far better now, though I still feel that the text is awkward and ambiguous in places. However, this is an editorial decision and I will not belabor the point here.

Experimental design

No comments

Validity of the findings

I am pleased to see that the author has taken on board nearly all the comments made in my review of the earlier version of this manuscript. The result is vastly improved and I have no further comments.

Additional comments

Just a note in passing: the number of variables in a DFA can certainly exceed the number of samples; however, it cannot exceed the number of cases within the smallest sample.

---

## Round 0.3 · accepted · Accept

Thank you for integrating these final suggestions and the reviewers for all their constructive comments. It was an honour handling this manuscript.